# Endoplasmic reticulum visits highly active spines and prevents runaway potentiation of synapses

Alberto Perez-Alvarez [1✉], Shuting Yin[1], Christian Schulze[1], John A. Hammer[2], Wolfgang Wagner [3] & Thomas G. Oertner [1✉]

In hippocampal pyramidal cells, a small subset of dendritic spines contain endoplasmic reticulum (ER). In large spines, ER frequently forms a spine apparatus, while smaller spines contain just a single tubule of smooth ER. Here we show that the ER visits dendritic spines in a non-random manner, targeting spines during periods of high synaptic activity. When we blocked ER motility using a dominant negative approach against myosin V, spine synapses became stronger compared to controls. We were not able to further potentiate these maxed-out synapses, but long-term depression (LTD) was readily induced by low-frequency stimulation. We conclude that the brief ER visits to active spines have the important function of preventing runaway potentiation of individual spine synapses, keeping most of them at an intermediate strength level from which both long-term potentiation (LTP) and LTD are possible.

---

[1] Institute for Synaptic Plasticity, Center for Molecular Neurobiology Hamburg (ZMNH), University Medical Center Hamburg-Eppendorf, Hamburg, Germany. [2] National Health, Lung and Blood Institute (NIH), Bethesda, MD, USA. [3] Institute for Molecular Neurogenetics, Center for Molecular Neurobiology Hamburg (ZMNH), University Medical Center Hamburg-Eppendorf, Hamburg, Germany. ✉email: alberto.perez-alvarez@zmnh.uni-hamburg.de; thomas.oertner@zmnh.uni-hamburg.de

The endoplasmic reticulum (ER) is a tubular network that pervades the entire neuron, including the full length of the axon[1] and all of its dendritic branches, reaching even some spines[2]. In addition to its canonical function in the synthesis and delivery of proteins and lipids, it is also an intracellular signaling system, as it is capable of buffering and releasing calcium ions into the cytoplasm[3]. ER membranes contact those of mitochondria, endosomes and also the plasma membrane for subcellular trafficking of lipids and calcium[4]. Fine ER tubules form sheets and cisternae, which run uninterrupted along the axon, supporting vesicle release at single boutons[5]. When proteins involved in molecular shaping of the ER are mutated, neurodegenerative processes are triggered[6]. In hippocampal CA1 pyramidal cells, EM studies have shown that only a small fraction of dendritic spines contain ER[7–9]. Functionally, spines containing ER express different forms of synaptic depression compared to ER-lacking spines on the same dendrite[10]. In large spines, ER forms a specialized organelle, the spine apparatus[7,9], which is readily identified by the presence of synaptopodin. Synaptopodin, an actin-associated protein originally discovered in renal podocytes[11], is also associated with the cisternal organelle inside the axon initial segment[12]. In Purkinje neurons, myosin Va (MyoVa), an actin-based motor, drives smooth ER tubules into virtually all spines during development[13], but no spine apparatus is formed in these neurons[11]. Synaptopodin knock-out mice are viable with relatively mild learning deficits[14]. The regulation and functional role of ER dynamics in dendritic spines remains unclear.

Here we investigate the dynamics of spine ER in CA1 pyramidal neurons in organotypic slice cultures of rat hippocampus. We show that ER is highly mobile, transiently entering most dendritic spines over time and persisting in a minority of spines. The frequency of spine entry events increased when synapses were active and ER motility was blocked by a MyoVa-based dominant-negative construct. Blocking ER motility in individual neurons led to strengthening of synapses and prevented further potentiation by a long-term potentiation (LTP) protocol. Long-term depression (LTD), on the other hand, was enhanced in neurons with blocked ER motility. Our findings support the concept that ER visits to spines are not random, but rather target spines with highly active synapses. Functionally, transient ER visits appear to limit runaway potentiation of these synapses.

## Results

**Time-lapse imaging of ER dynamics**. We assessed the presence of ER in dendritic spines of CA1 pyramidal cells expressing the red fluorescent protein tdimer2 in the cytoplasm and EGFP in the ER (Fig. 1a). To quantify ER dynamics, we imaged oblique dendrites in stratum radiatum at 10 min intervals with a two-photon microscope (Fig. 1b, c). About 20% of dendritic spines contained ER at any single time point (Fig. 1d), consistent with previous reports[7,10]. Imaging for 5 h, the majority of spines (71%) were visited by ER at least once (Fig. 1c, f). ER visits were typically short, often appearing at single time points (Fig. 1e, Supplementary Movie 1). We also observed spines (~10%) which contained ER during the entire observation period (Fig. 1b, c, f). We hypothesized that these stably ER-positive spines contained a spine apparatus, and using 3D image stacks (Supplementary Fig. 1), we could indeed confirm synaptopodin immunoreactivity in 90% of these spines. In contrast, only 16% of spines with transient ER visits were scored as synaptopodin-positive (Fig. 1g, h). Within this group, spines that were scored ER-positive right before fixation were more likely to contain synaptopodin (20%, Supplementary Fig. 2). Four percent of the spines that were never visited by ER stained positive against synaptopodin (Fig. 1g, h), which could indicate accumulation of this soluble protein prior to

ER visits[15]. Strong synaptopodin immunoreactivity was also seen in dendritic shafts and at the axon initial segment, as reported previously[12,15,16].

Intrigued by the highly dynamic nature of ER visits to spines, we tested the role of excitatory synaptic transmission in this process. Blocking AMPA and NMDA receptors reduced the proportion of spines transiently visited by ER from 37% (control) to 15%, also decreasing the total time the organelle spent in spines (Fig. 1i, j). In contrast, blocking group I mGluRs with a cocktail of MPEP and LY367385 strongly increased the proportion of transient ER spines to 65% and prolonged the duration of visits (Supplementary Fig. 3). Thus, on a global level, ER motility is boosted and spine visits are prolonged by fast excitatory transmission, while mGluR activation counteracts these effects, consistent with an earlier study on dissociated hippocampal neurons[17].

**Induction of structural LTP at spine synapses**. To test whether activation of glutamate receptors on a single spine is sufficient to attract ER to that spine, we induced structural LTP by two-photon uncaging of MNI-glutamate in $Mg^{2+}$-free saline and TTX. This approach circumvents the presynaptic terminal and leads to maximal activation of postsynaptic NMDA receptors[18]. Uncaging induced a lasting volume increase of spines that had no ER at baseline, but had no consistent effect on ER+ spines (Fig. 2a, b). This difference in structural plasticity could be due to efficient removal of free $Ca^{2+}$ by SERCA pumps[19] in ER+ spines. In eight out of ten spines showing structural LTP, ER was immediately attracted into the stimulated spine (Fig. 2c, d; Supplementary Movie 2), but never to neighboring, nonstimulated spines. Thus, strong synaptic activity, typically associated with synaptic potentiation, triggers ER visits.

Next, we asked whether "spontaneous" ER visits to spines were perhaps also triggered by high activity of the resident synapse. Since calcium influx into spines as a result of synaptic activation triggers actin polymerization and expansion of the spine head[20], volume fluctuations of individual spines provide some information about the activity of the impinging synapse. We tracked spine head volume changes for several hours to look for any correlation with ER visits. We observed spine enlargement of variable intensity and duration preceding the ER visit, followed by rapid collapse back to baseline volume after ER retraction (Fig. 2e). As ER visits occurred at different time points, the average fluorescence intensity across all monitored synapses was constant over the course of the experiment (Fig. 2f, left). When we aligned the spine head volume traces (tdimer2 fluorescence intensity normalized to the local dendrite) of all recorded spines to the time point when they reached their peak volume, we found that at the time the spine head was largest ($t = 0$), the probability of ER entry was maximal (Fig. 2f, center). This is consistent with the idea that ER preferentially visits a spine after the resident synapse was highly active. Vice versa, aligning ER entry of all spines to a single time point confirmed that spine volume reached its maximum at ER insertion. Thus, although there might have been strong synaptic activity that led to spine swelling, there was no sign of lasting structural plasticity following the "spontaneous" ER visit (Fig. 2f, right).

**ER dynamics is driven by myosin motors**. To explore whether ER visits affect synaptic function we needed to interfere with ER dynamics. In Purkinje cells of the cerebellum, it is well established that MyoVa pulls ER into spines where it is subsequently retained[13]. Therefore, to attenuate myosin V-based ER transport, we expressed MyoV DN, a dominant-negative construct comprising the MyoVa globular tail domain (GTD) fused with a

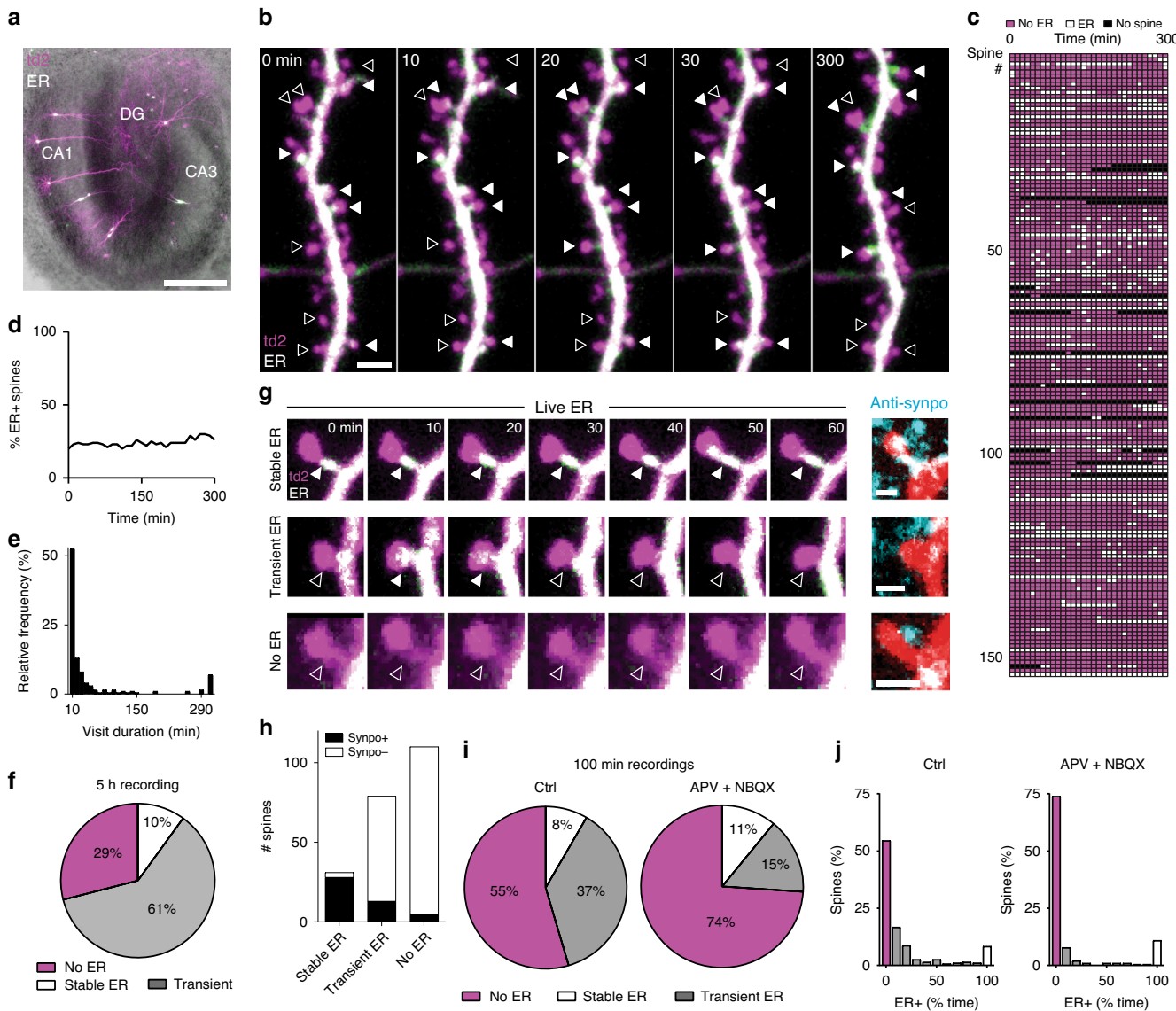

**Fig. 1 ER dynamics is regulated by synaptic activity. a** Organotypic hippocampal slice transfected using a Helios gene gun. Overlay of bright field and fluorescence images shows expression of tdimer2 (magenta) and ER-EGFP (green fluorescence, white in overlay) in a few CA1 neurons. Scale bar: 500 μm. Experiment was reproduced in ten slices. **b** Two-photon maximum intensity projections of a dendritic branch from a hippocampal CA1 pyramidal neuron transfected with tdimer2 (magenta) and ER-EGFP (green, white when over magenta) followed at 10 min intervals up to 5 h. Filled and empty arrowheads denote presence and absence of ER in a few representative spines, respectively. Scale bar: 2 μm. Horizontal structure is an axon. Experiment was reproduced on three slices with similar results. **c** Score sheet of dendritic spines monitored over 5 h (n = 153 spines, 2 neurons, 2 slices). Magenta: spine without ER; white: spine with ER; black: spine head not visible/not analyzed. **d** Percentage of spines from **c** containing ER over time. **e** Histogram of ER visit duration over a 5 h period (from **c**). **f** Percentage of spines with stable ER, receiving at least one visit (transient ER) or no visit (no ER) within 5 h of observation (from **c**). **g** Three examples (two-photon time series, maximum intensity projections) of spines with stable (top), transient (middle), or no ER (bottom), followed by correlative confocal images (maximum intensity projections) of the same spines (native tdimer2 fluorescence, red) after fixation of the tissue and immunostaining against synaptopodin (cyan). Scale bars: 1 μm. Note synaptopodin clusters inside (white) as well as outside (cyan) the transfected neuron (red), as the antibody labels synaptopodin in the entire neuropil. Experiment was reproduced on two slices. **h** Synaptopodin clusters were detected in 90% of stable ER+ spines, in 16% of transient ER spines, and in 4% of spines with no ER visit (n = 220 correlatively imaged spines, 2 cells, 2 slices). **i** Over a period of 100 min, 45% of spines were visited by ER at least once. Blocking excitatory transmission (APV + NBQX) reduced the fraction of transiently visited spines from 37% (n = 262 spines, 3 neurons, 3 slices) to 15% (n = 192 spines, 2 neurons, 2 slices). **j** Residence time of ER in spines, under control conditions (n = 262 spines, 3 neurons, 3 slices) and with blocked excitatory transmission (n = 192 spines, 2 neurons, 2 slices) (p = 0.0005, two-sided Mann–Whitney test). Source data are provided as a Source Data file.

dimerizing leucine zipper and known to reduce ER targeting to Purkinje neuron spines[15]. Using CA1 pyramidal cells expressing a leucine-zipper-fused fluorescent label (mCerulean-LZ) as a control, we analyzed neuronal ER while blind to the genotype. Expression of MyoV DN in CA1 neurons decreased fivefold the proportion of spines that contained ER, from 20 to 4% (Fig. 3a–c, Supplementary Fig. 4), while the density of spines on the dendrite was similar in both groups (Fig. 3d). Transient ER visits were extremely rare in MyoV DN neurons, decreasing from 41 to 3% (Fig. 3e, Supplementary Movie 3). A small number of spines (2%)

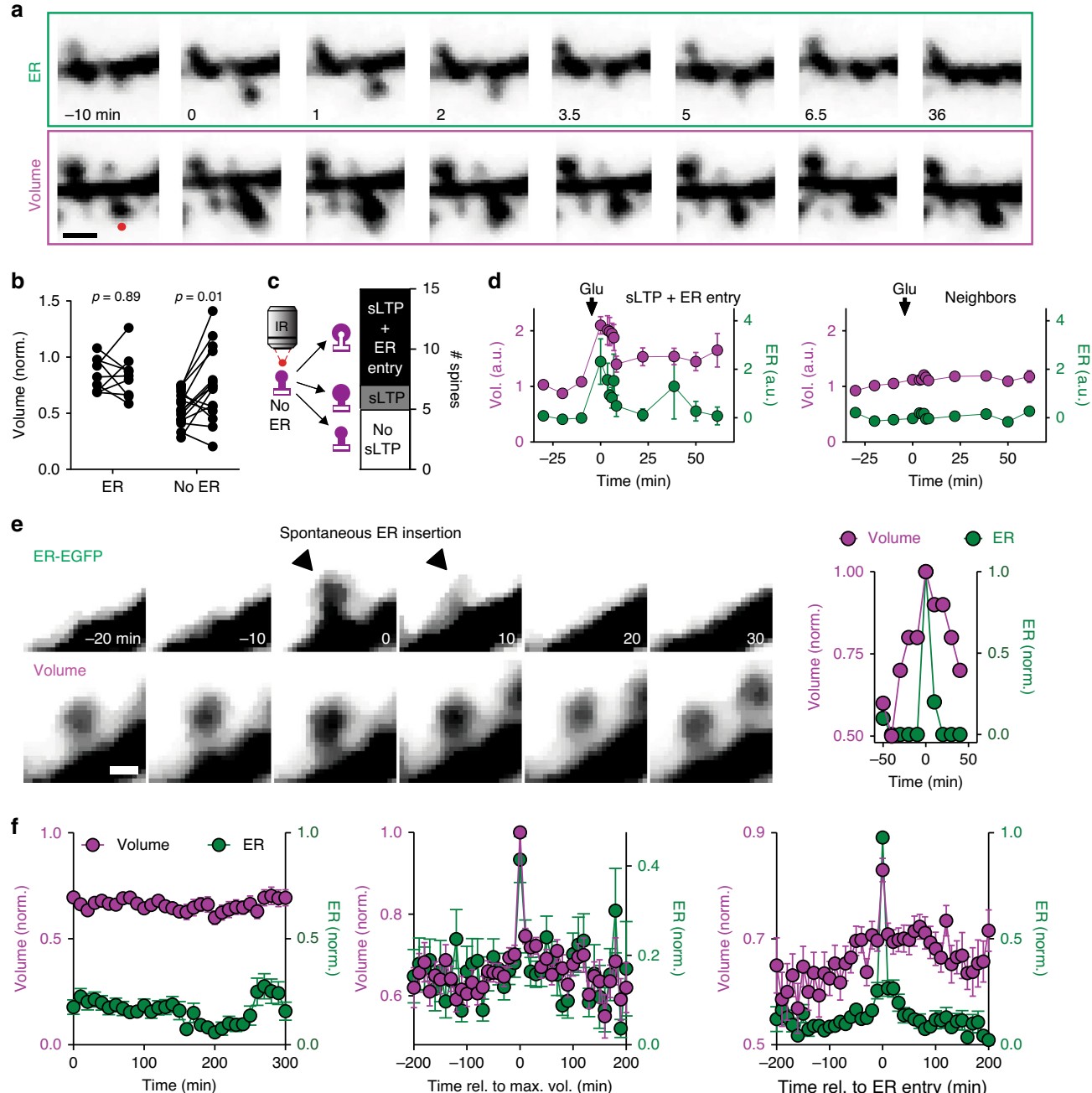

**Fig. 2 Spine structural plasticity triggers ER visits. a** Time-lapse imaging of tdimer2 (spine volume) and ER-EGFP fluorescence during a glutamate uncaging experiment in zero Mg²⁺. Red dot denotes location of uncaging spot immediately before $t = 0$. Scale bar: 1 μm. Experiment was reproduced eight times. **b** Size of spines (normalized to local dendrite) before and after glutamate uncaging. Spines without ER were smaller ($0.55 \pm 0.04$, $n = 15$ spines, 3 cells, 3 slices) than those containing ER ($0.84 \pm 0.05$, $n = 9$ spines, 3 cells, 3 slices) ($p = 0.0001$, two-sided unpaired $t$-test). Spines without ER responded with a volume increase to uncaging ($0.75 \pm 0.08$, $n = 15$, $p = 0.01$, two-sided paired $t$-test), while ER spines remained largely unchanged ($0.83 \pm 0.07$, $n = 9$, $p = 0.89$, two-sided paired $t$-test). **c** Lasting volume increase (sLTP, ten spines) was typically associated with ER entry (eight of ten spines). Spines that did not show sLTP (five spines) also did not attract ER. **d** Average volume and ER signal (normalized to pre-uncaging) of stimulated spines undergoing structural LTP and ER insertion ($n = 8$ spines) and immediate neighbors ($n = 39$ spines) upon glutamate uncaging. Arrows indicate time of glutamate uncaging. Markers and error bars represent mean ± SEM. **e** Example of a spine receiving a brief ER visit during a volume maximum. Scale bar: 1 μm. This behavior was reproduced on spines from three slices. **f** Averaging spine volume traces over the entire observation period shows stable average volume and stable probability of ER visits (left). Aligning spine volume traces by the time of maximum spine volume ($t = 0$) reveals coincident ER entry, but no lasting structural plasticity following the entry event (middle). Aligning traces by maximum ER signal upon entry reveals simultaneous spine head volume maximum (right). Prior to averaging, fluorescence intensity over time was normalized to the maximum fluorescence measured in each spine. Datapoints are mean ± SEM from $n = 65$ spines (3 slices). Source data are provided as a Source Data file.

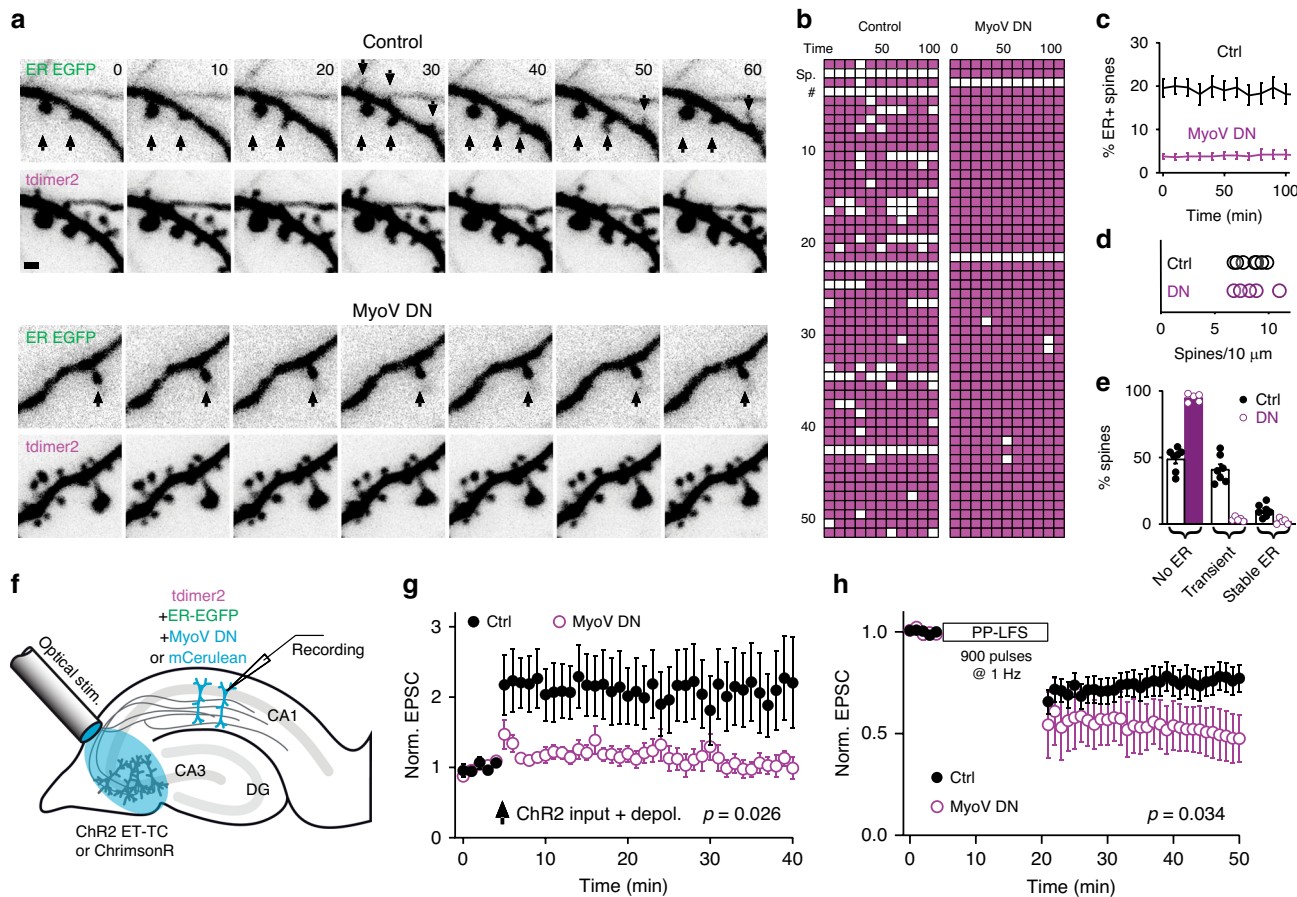

**Fig. 3 ER motility is driven by myosin V. a** Time-lapse imaging of ER motility in dendritic sections of CA1 pyramidal cells expressing just the fluorescent markers (control) or, in addition, the dominant-negative construct MyoV DN. Scale bar: 1 μm. **b** Spine ER score charts of a MyoVa DN-expressing neuron and control (ER+ spines: white; ER− spines: magenta). Note persistent ER in two spines of the MyoVa DN neuron. **c** Percentage of spines containing ER over time in control ($n = 896$ spines, 7 slices, 7 neurons) and MyoV DN-expressing CA1 neurons ($n = 567$ spines, 5 slices, 5 neurons). Data are mean ± SEM. **d** Spine densities on dendrites of control (seven slices, seven neurons) and MyoV DN neurons (five slices, five neurons) (8.3 ± 0.4 vs 8.4 ± 0.7 spines per 10 μm, $p = 0.8835$, two-sided unpaired $t$-test). Points show mean ± SEM. **e** MyoV DN significantly increased the number of spines that were never visited by ER (49 ± 3% vs 95 ± 1%, $p < 0.0001$, unpaired $t$-test), decreased the number of transient ER visits (41 ± 4% vs 4 ± 1%, $p < 0.0001$, two-sided unpaired $t$-test) and the number of spines containing stable ER (10 ± 2% vs 2 ± 1%, $p = 0.0052$, two-sided unpaired $t$-test) over a 100 min imaging period. Markers represent individual neurons ($n = 7$ neurons for control, $n = 5$ neurons for MyoV DN), bars show mean ± SEM. **f** Pairing optogenetic stimulation of CA3 pyramidal cells with step depolarization of a patch-clamped CA1 pyramidal cell expressing fluorescent proteins and MyoV DN or just fluorescent proteins (control). **g** Pairing optogenetic stimulation with 100 ms postsynaptic depolarization to −15 mV (ten repeats) induced LTP in control CA1 pyramidal cells, but not in neurons expressing MyoV DN instead of mCerulean (ctrl = 221% ± 55% of baseline, $n = 10$ neurons, 10 slices; MyoV DN = 97% ± 11% of baseline, $n = 12$ neurons, 12 slices; $p = 0.026$, two-sided unpaired $t$-test). Datapoints are mean ± SEM. **h** Low-frequency stimulation (900 repeats, 1 Hz, paired pulses) induces stronger LTD in neurons expressing MyoV DN (ctrl: 77% ± 7% of baseline, $n = 12$ neurons, 12 slices; MyoV DN: 47% ± 12% of baseline, $n = 5$ neurons, 5 slices; $p = 0.033$, two-sided unpaired $t$-test). Datapoints are mean ± SEM; source data are provided as a Source Data file.

still had stable ER in MyoV DN-expressing neurons, suggesting that spines with already stable presence of ER, likely containing a spine apparatus, did not require MyoVa for its maintenance (Fig. 3e, Supplementary Fig. 4). Overexpression of functional MyoVa doubled the fraction of ER+ spines and counteracted the effects of the DN construct (Supplementary Fig. 5), suggesting that the rate of ER visits is limited by the availability of MyoVa. After establishing that MyoV DN altered ER motility, we investigated whether synaptic plasticity was also affected. Pairing presynaptic stimulation with postsynaptic depolarization[21] induced LTP in control CA1 pyramidal cells, but not in cells expressing MyoV DN (Fig. 3f, g). In contrast, mGluR-dependent LTD (mGluR LTD), a form of hippocampal LTD[22], was enhanced in MyoV DN-expressing CA1 pyramidal cells (52% decrease in EPSC amplitude) compared to control neurons (23% decrease in EPSC amplitude, Fig. 3h).

**Block of ER visits induces runaway potentiation of synapses.** Electrophysiological induction of long-term plasticity assesses relative changes in synaptic strength, but provides little information about the absolute strength of synapses at baseline, e.g. the number of glutamate receptors. To measure surface expression of AMPA receptors at individual synapses, we expressed GluA2 subunits fluorescently tagged with super-ecliptic pHluorin (SEP-GluA2)[23,24]. Five days after electroporation, GluA2 surface expression was higher on spines of MyoV DN-expressing neurons than on spines of control neurons (Fig. 4a, b, Supplementary Fig. 6). This GluA2 enrichment was not associated with an increase in spine volume, as the distribution of spine head volumes was not different between MyoV DN-expressing and control neurons. Accordingly, the GluA2 concentration on spines was higher in MyoV DN neurons (Fig. 4b). This prompted us to measure the potency of

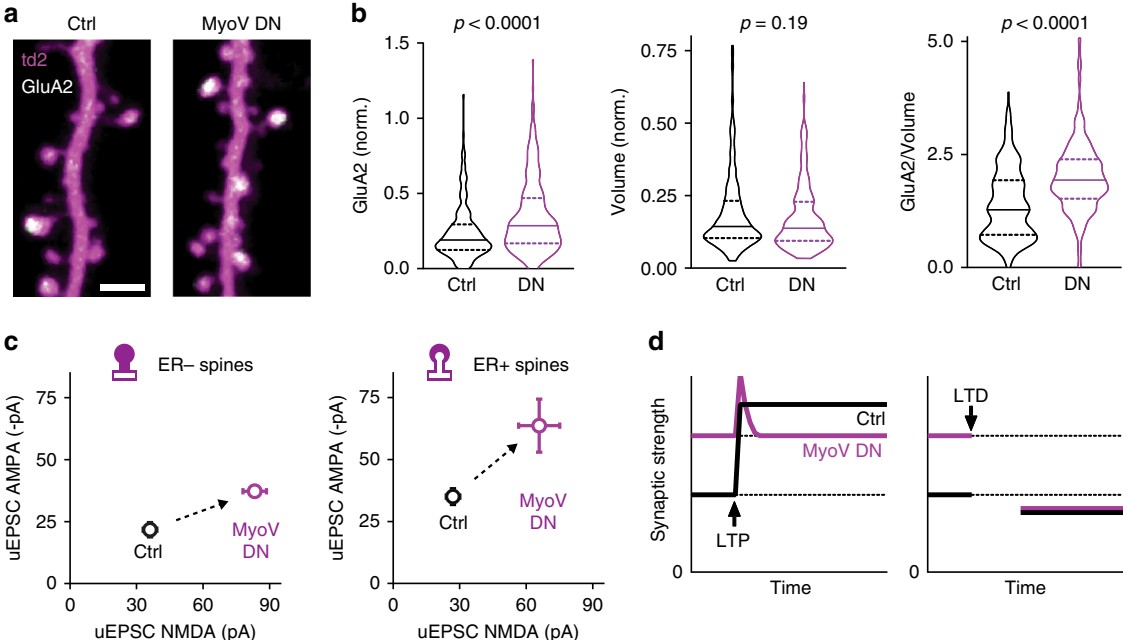

**Fig. 4 Blocking ER motility leads to high potency synapses. a** Two-photon images showing distribution of SEP-GluA2 receptors in dendritic spines of control and MyoV DN-expressing CA1 neurons. Scale bar: 2 μm. This experiment was repeated in 4 slice cultures. **b** Violin plots showing median (solid horizontal line) and quartiles (dashed horizontal lines) of SEP-GluA2 content (median 0.19 vs 0.29, $p < 0.0001$, Mann–Whitney test), spine head volume (0.143 vs 0.137, $p = 0.2$, two-sided Mann–Whitney test), and SEP-GluA2 concentration (1.28 vs 1.93, $p < 0.0001$, two-sided Mann–Whitney test) in dendritic spines of control ($n = 266$ spines, 4 neurons, 2 slices) and MyoV DN-expressing CA1 neurons ($n = 314$, 4 neurons, 2 slices). **c** Uncaging-evoked AMPA-EPSCs recorded at −70 mV were larger at synapses of MyoV DN-expressing neurons (Ctrl ER− 21.7 ± 2.9 pA, $n = 16$ spines vs MyoV DN ER− 37.2 ± 2.5 pA, $n = 58$ spines, $p = 0.0009$, two-sided Mann–Whitney test; Ctrl ER + 35.0 ± 3.2 pA, $n = 17$ spines vs MyoV DN ER + 63.6 ± 10.7 pA, $n = 10$ spines, $p = 0.0018$, two-sided Mann–Whitney test). Uncaging-evoked NMDA-EPSCs recorded at +40 mV were also larger at synapses of MyoV DN-expressing neurons (Ctrl ER− 36.0 ± 3.1 pA, $n = 16$ spines vs MyoV DN ER− 83.2 ± 5.3 pA, $n = 58$ spines, $p < 0.0001$, two-sided Mann–Whitney test; Ctrl ER + 26.9 ± 3.0 pA, $n = 17$ spines vs MyoV DN ER + 65.8 ± 9.3 pA, $n = 10$ spines, $p < 0.0001$, two-sided Mann–Whitney test). Markers show mean ± SEM. **d** Combining the information about baseline strength (uEPSC AMPA) and long-term plasticity suggests that MyoV DN expression did not change the dynamic range of synapses. Source data are provided as a Source Data file.

individual synapses in neurons that did not overexpress AMPAR subunits. To measure native AMPAR and NMDAR currents at individual spines, we used two-photon uncaging of glutamate in standard artificial cerebrospinal fluid (ACSF) at two different holding potentials, −70 and +40 mV (Fig. 4c). In control neurons with intact ER motility, we confirmed our previous report[10] that spines with ER contain almost twice as many AMPA receptors than those devoid of ER. In neurons where ER motility was blocked by MyoV DN, glutamate uncaging on individual spines produced significantly larger AMPAR and NMDAR currents (Fig. 4c). The strong effect of MyoV DN expression on baseline potency suggests that is misleading to present plasticity experiments (Fig. 3g, h) normalized to a 100% baseline. When the different AMPAR potencies at baseline are taken into consideration, the lack of LTP and the enhanced LTD in MyoV DN neurons is consistent with a dynamic range very similar to the synapses of control neurons (Fig. 4d).

## Discussion
Our study addresses two questions: (1) Are ER visits to dendritic spines random events? (2) What is the consequence of ER presence for synaptic plasticity? With regard to the first question, we present three lines of evidence suggesting that ER selectively visits spines during periods of high synaptic activity. First, blocking synaptic transmission in spontaneously active slice cultures strongly reduces ER dynamics. Second, spines are preferentially visited by ER during phases of rapid volume expansion, as typically accompanies LTP[25]. Third, ER entry can be externally

triggered by glutamate uncaging at individual spines. Thus, transient ER visits are not random, but directed to spines with active synapses.

We show that block of myosin V by our DN construct strongly reduces the number of ER visits to CA1 pyramidal cell spines, consistent with the known role of this motor protein in driving ER tubules into Purkinje cell spines[13]. High calcium levels enhance cargo binding of myosin V[26]. In addition, the burst of actin polymerization known to drive LTP-associated spine expansion also provides myosin V motors with the right tracks (ADP-Pi rich, barbed end out) to pull ER tubules from the dendrite through the narrow neck into expanding spines[27]. We therefore propose that the nonrandom ER visits are to spines that have just undergone an LTP-like event. This interpretation is supported by studies of synaptic ultrastructure: following LTP-inducing stimulation in acute hippocampal slices, the fraction of large spines with large PSDs increases, and many of these large spines contain ER, often in the form of a spine apparatus[28,29]. When we used 2P glutamate uncaging to induce structural LTP at single spines, we transiently attracted ER into the swelling spines and did not observe stabilization of spine ER that would suggest formation of a spine apparatus. Possibly multiple synapses need to simultaneously be potentiated (as with theta-burst stimulation) for the redistribution of dendritic ER towards potentiated spines, which may be necessary to provide material for spine apparatus formation[29]. Alternatively, de novo spine apparatus assembly might be easier to trigger in the acute slice preparation, where most spines have been newly formed in the hours after slice making[30].

In our study, spines with transient ER visits were mostly synaptopodin-negative. This is not surprising, as the process of spine apparatus assembly inside a spine takes about 1 h[15]. Indeed, we detected synaptopodin in 90% of spines with stable ER, and after several days of MyoV DN expression, the fraction of these stable ER spines was halved. This suggests that once a spine apparatus is assembled and sufficiently stabilized by synaptopodin, myosin activity is not constantly required for its maintenance[15]. The situation might be different in Purkinje cells, which do not express synaptopodin, do not form spine apparatus[14], but still are able to sustain tubular ER in every spine[13]. Structurally, Purkinje cell spines are much less dynamic than CA1 pyramidal cell spines and regulate actin organization in a distinct manner[31]. Therefore, organization of the actin tracks, not the processive motor protein itself, may be responsible for the differences in spine ER stability.

As for the potential function of spine ER, we report that in neurons with compromised ER motility, spine synapses have increased GluA2 and larger glutamate responses. These strengthened synapses are not further potentiated by an LTP protocol, but are strongly depressed by low-frequency stimulation. Similarly, in cortical pyramidal cells of flailer mice, which express an intrinsic dominant-negative MyoVa fragment and lack spine ER in their Purkinje cells[32], both the amplitude and frequency of spontaneous AMPAR currents are strongly enhanced[33]. A pioneering study expressing a different MyoVa-GTD construct in CA1 neurons also reported GluA2 enrichment in spines and blocked LTP, but in contrast to our results, AMPAR currents were reduced 15 h after transduction[34]. Presence of spine ER was not assessed by Correia et al.[34], but in our hands, a nondimerizing GTD as they used does not affect ER spine targeting[15]. Thus, it is possible that the potentiation of baseline strength we report here happens only when spine synapses are not visited by ER for several days, or it reflects an increase in extrasynaptic receptors.

When comparing results of dominant-negative interference with myosin V on synaptic strength and plasticity, it is important to point out that MyoVa has cargo-binding sites on both its GTD and its stalk (coiled-coil region). Our DN construct with leucine zipper dimerization was designed to compete with intact MyoVa specifically for its GTD-binding cargo since it lacks the MyoVa coiled coil[15]. As we expected, expression of MyoV DN blocked ER motility almost completely, but it may also have affected the distribution of other GTD-binding cargos. Of obvious importance for synaptic plasticity is the GluA1 subunit of the AMPA receptor[34], which may be transported by MyoVa or MyoVb isoforms[26,35]. Interestingly, hippocampal LTP is intact in the MyoVa null mutant dilute-lethal, suggesting that both MyoV isoforms may be able to traffic GluA1[36]. While disturbed GluA1 delivery likely contributed to the lack of LTP we observed after MyoV DN expression, it does not explain why both AMPA and NMDA responses grew stronger and spine surface expression of GluA2 was enhanced.

Exocytosis and transport of dense-core vesicles is increased by a MyoVa DN[37], possibly enhancing release of BDNF, an important regulator of synaptic strength. BDNF release from spines is necessary for LTP induction[38] and might contribute to the increased baseline synaptic strength in MyoV DN neurons. However, BDNF release from dendrites and spines is unlikely to be from dense-core vesicles and the mechanism has not been determined.

There are other MyoVa binding proteins affecting synaptic plasticity such as GKAP, a scaffold protein accumulating in spines in an inverse relation to activity and CaMKIIα[39]. Overexpression of a nonfunctional GKAP mutant impaired homeostatic synaptic scaling in both directions, but had no effect on the frequency or amplitude of spontaneous synaptic currents[40]. GKAP is thought to bind to the myosin stalk[41], so we would not expect it to be affected by our DN construct.

While we acknowledge the limited specificity of the DN approach[35] and the naiveté of monocausal explanations, it is difficult to imagine how disruption of cargo transport or anchoring would lead to an upwards drift of synaptic strength unless the cargo is a negative regulator of synaptic strength. Previously, we observed that only ER-containing spines are able to undergo mGluR-dependent LTD[10]. At that time, we were surprised how few hippocampal spines were competent to express LTD. Now, taking ER dynamics into account, we observe that within a 5 h window, most spines are visited by ER at least once and that these transient ER visits coincide with LTP-like events. In combination with the MyoV DN observations, we propose that in hippocampal neurons, ER visits strongly activated spines and limits potentiation by a group1 mGluR-dependent mechanism.

While the importance of ER-mediated, spine-specific depression or depotentiation of Schaffer collateral synapses has not been recognized, there is ample evidence for the existence of this mechanism in the amygdala[42] and in the cerebellum[43]. In Purkinje cells, where each spine contains ER and IP$_3$ receptors, LTD is strongly dependent on IP$_3$ signaling[44]. MyoVa-null mutants lack ER and IP$_3$ receptors in Purkinje cell spines and consequently, show impaired LTD[44]. IP$_3$ generated locally at these spines produces a small delayed calcium transient, not sufficient to trigger LTD, which can however be rescued by simultaneous calcium uncaging[44]. Other mutants endowed with very low levels of functional MyoVa (dilute-neurological) are able to recover LTD and motor learning in the adult stage, just when MyoVa expression is sufficient to translocate ER tubules into a few spines[45]. In CA1 pyramidal cells, local activation of mGluRs on ER-containing spines triggers calcium release via activation of IP$_3$ receptors[10]. Thus, is plausible that chronically removing this brake mechanism, by blocking active ER entry into spines, leads to runaway potentiation of CA1 synapses. Interestingly, blocking mGlu1 and mGlu5 dramatically increased spine ER visits whereas DHPG reduces visits[46], perhaps as a homeostatic response to disruption of the LTD pathway. But if LTD requires ER in spines, how could LTD be increased in MyoV DN neurons? A possibility is that, in addition to IP$_3$ receptor activation, CA1 neurons possess an alternative signaling pathway for mGluR-mediated LTD: mGluRs activate protein tyrosine phosphatases, resulting in dephosphorylation of AMPARs and in consequence, their removal from the synapse[47]. This pathway, which does not depend on the presence of ER, may be responsible for mGluR-mediated LTD in MyoV DN CA1 neurons.

Our model of ER-mediated depotentiation raises the question how successful LTP is possible under physiological conditions with fully functional ER. It has been shown that neuromodulators such as dopamine or acetylcholine promote plasticity by blocking SK channels, calcium-activated K$^+$ channels that normally limit spine head depolarization[48,49]. The resulting very strong NMDA currents might be able to override the homeostatic function of spine ER, leading to successful induction of LTP. This situation is mimicked by glutamate uncaging in zero Mg$^{2+}$ saline, which also maximizes Ca$^{2+}$ influx into the spine and provides an extremely strong stimulus. Under these conditions, we could induce lasting structural LTP in spite of ER entry.

In comparing our results to the literature, it is important to point out that our interference with myosin V driven cargo transport was cell-specific, while most published studies used pharmacological or global genetic interventions that blocked motor function in all neurons. Myosin V has important presynaptic functions[50], but presynaptic neurons were not affected in our experiments. This allowed us to assign the observed changes

in synaptic function and plasticity unequivocally to altered signaling in the postsynaptic neuron.

## Methods

**Constructs and transfection of hippocampal organotypic cultures**. Hippocampal slice cultures from Wistar rats (both sexes) were prepared at postnatal day 4–5[51]. No antibiotics were added to the culture medium. Animal procedures were in accordance with the guidelines of local authorities and Directive 2010/63/EU. All protocols were approved by the Behörde für Gesundheit und Verbraucherschutz of the City of Hamburg. At DIV 7, slice cultures were biolistically transfected with pCI-hsyn-tdimer2[52] and pMH4-hsyn-ER-EGFP plasmids (RRID:Addgene_22285) using a Helios Gene Gun (Bio-Rad) and imaged 2–3 weeks after. For immunohistochemistry and expression of more than two constructs, we electroporated individual CA1 neurons at DIV 21 and imaged 3–14 days later. Briefly, thin-walled pipettes (~10 MΩ) were filled with intracellular K-gluconate based solution into which plasmid DNA was diluted to 20 ng μl$^{-1}$. The intracellular solution contained in (mM): 135 K-gluconate, 4 MgCl$_2$, 4 Na$_2$-ATP, 0.4 Na-GTP, 10 Na$_2$-phosphocreatine, 3 sodium-L-ascorbate, 0.02 Alexa Fluor 594, and 10 HEPES (pH 7.2). Pipettes were positioned against neurons and DNA was injected using an Axoporator 800A (Molecular Devices) with 50 hyperpolarizing pulses (−12 V, 0.5 ms) at 50 Hz. To enable blind experiments and analysis, DNA mixes were coded by a second researcher and only after all recordings and analysis were completed was the investigator unblinded. To generate MyoV DN, a sequence encoding the GTD of mouse MYO5A (starting at residue 1415, numbering according to brain-spliced isoform) was inserted in frame at the 3′-end of the leucine zipper of mCer-LZ. Plasmid mCer-LZ was generated by inserting a sequence encoding the leucine zipper of GCN4 (MKQLEDKVEELLSK NYHLENEVARLKKLVGE) in frame at the 3′-end of the mCerulean coding sequence[53]. When assessing the effect of MyoV DN, mCer-LZ was used in control cells in substitution of MyoV DN to maintain an identical DNA concentration in electroporation mixes and for post-hoc identification. Both constructs were inserted into pCI backbones under the human synapsin1 promoter.

**Two-photon imaging and scoring of ER+ spines**. The custom-built two-photon imaging setup was based on an Olympus BX51WI microscope equipped with a LUMPLFLN-W 1.1 NA collar-corrected Olympus objective, controlled by the open-source software package ScanImage[54]. A Ti:sapphire laser (MaiTai DeepSee, Spectra Physics) controlled by electro-optic modulators (350-80, Conoptics) tuned to 980 nm was used to simultaneously excite ER-EGFP and tdimer2. The point spread function (PSF) measured with 0.1 μm FluoSpheres (Thermo Fisher) was 0.35 × 0.35 × 1.5 μm. Z-stacks (0.3 μm steps) of expressing neurons were acquired at 10 min intervals and 37 °C. Emitted photons were collected through objective and Peltier-heated oil-immersion condenser (1.4 NA, Olympus) with two pairs of photomultiplier tubes (PMTs, H7422P-40, Hamamatsu), mounted on the cool side of three Peltier elements. 560 DXCR dichroic mirrors and 525/50 and 607/70 emission filters (Chroma Technology) were used to separate green and red fluorescence. Bleed-through of ER-EGFP fluorescence into the red channel was quantified expressing ER-EGFP alone in neurons and measuring the red fluorescence counts relative to the green fluorescence counts (15%) under 980 nm excitation. Green fluorescence of immature tdimer2 was quantified in cells expressing only tdimer2[52]. Excitation light was blocked by short-pass filters (ET700SP-2P, Chroma). Slice cultures were continuously superfused with a peristaltic pump (Gilson) and a single inline solution heater (Warner Instruments) with ACSF saturated with 95% O$_2$ and 5% CO$_2$ at 37 °C containing 127 mM NaCl, 25 mM NaHCO$_3$, 25 mM D-glucose, 2.5 mM KCl, 1 mM MgCl$_2$, 2 mM CaCl$_2$, and 1.25 mM NaH$_2$PO$_4$ (pH 7.4, 320 mOsm kg$^{-1}$). After a 15 min adaptation period, oblique dendrites from the proximal apical region of CA1 pyramidal neurons were selected for time-lapse imaging (3D image stacks every 10 min for 3–5 h).

To evaluate ER dynamics, maximum projections of individual Z-stacks were aligned using rigid registration in Fiji[55,56]. Individual spines were identified in the red (volume) channel and numerically annotated. In the green channel (ER), protrusions from the shaft and subsequent retractions were scored as ER entry and exit, respectively. In score charts of time-lapse experiments, white color denotes GFP-signal from the ER present inside a tdimer2-filled spine (green over magenta adds to white). At time points when a spine could not be resolved (i.e., retraction or proximity <0.5 μm to the dendrite axis), no ER scoring could be allocated (black color). If a spine appeared intermittently in the same location, it was scored in the same chart row. If ER entered at least once the dendritic spine head or neck, spines were scored as Transient ER. If ER was present in the spine throughout the experiment, it was assigned to the Stable ER group. Spines that never experienced ER entry were classified as No ER. Spine density was calculated dividing the number of visually detected spines by the length of the dendritic section measured in Imaris (Oxford Instruments).

For spine head fluorescence analysis in large time-lapse datasets, automatic detection of spines was performed with SpineChecker[57]. To correct for depth-dependent attenuation, fluorescence values at detected spines were normalized to local dendritic shaft fluorescence. Bleed-through of EGFP fluorescence into the red channel (15%) and tdimer2 fluorescence into the green channel (5%) was corrected. Thapsigargin experiments, which desynchronized ER entry and spine

volume maxima (Supplementary Fig. 7), show that the correlation between spine volume and ER presence was not caused by EGFP fluorescence detected in the red (volume) channel.

**Uncaging of MNI-glutamate to induce structural LTP**. MNI-glutamate (5 mM), TTX (1 μM), and D-Serine (30 μM) were added to ACSF containing zero Mg$^{2+}$ and 3 mM Ca$^{2+}$. All compounds were purchased from Tocris. To obtain a baseline of spine volume and ER content, two-photon image stacks covering a small stretch of dendrite were acquired at 980 nm excitation. To induce sLTP, glutamate was uncaged by a series of 60 laser pulses (720 nm, 1 ms, 1 Hz) delivered to the edge of a spine furthest from the dendritic shaft[18]. Spine volume and ER fluorescence (tdimer2; ER-EGFP) was monitored at regular intervals up to 100 min after the uncaging protocol.

Morphological analysis was performed using custom written software (MATLAB, MathWorks). For a relative measure of spine volume, spine fluorescence was normalized to local dendritic fluorescence. Briefly, a circular ROI was centered on the spine head at the time point where the uncaged spine reached its largest volume. Within the ROI, the brightest 20% of pixels were averaged in every z-plane. The maximum fluorescence, corresponding to the z-plane intersecting the center of the spine, was selected and normalized to the local dendritic fluorescence.

**Uncaging of MNI-glutamate to measure synaptic potency**. The potency of individual synapses was measured by uncaging of MNI-glutamate (720 nm, 1 ms pulses, five repeats at 0.1 Hz) on individual spines in ACSF 4/4 (see below) at room temperature (RT). Postsynaptic neurons were voltage-clamped, first at −70 mV to measure AMPAR uEPSPCs and then at +40 mV to measure NMDAR uEPSCs, using a pipette solution that contained (in mM) 135 Cs-MeSO$_4$, 4 MgCl$_2$, 4 Na$_2$-ATP, 0.4 Na-GTP, 10 Na$_2$-phosphocreatine, 3 ascorbate, and 10 HEPES (pH 7.2, 295 mOsm kg$^{-1}$).

For both control and MyoV DN groups, uncaged spines were located on thin oblique dendrites at a distance equal or below 100 μm from the somatic recording pipette. Due to the high electrical resistance of spine necks, voltage clamp of spine synapses is far from perfect[58,59]. Thus, the true synaptic conductance may be higher than our measured EPSCs suggest. However, as the diffusional coupling of ER-negative and ER-positive spines is similar[10], we have no reason to suspect systematic differences in spine neck resistance between control and MyoV DN groups. Thus, the large difference in EPSC amplitude most likely reflects a proportional increase in the number of glutamate receptors per spine in MyoV DN-expressing neurons.

**Electrophysiology and plasticity induction**. Two to three weeks before recording, hippocampal slice cultures were microinjected in the CA3 area with an adeno-associated virus (AAV7 or AAV9, prepared at the UKE vector facility) to express either ChR2 ET-TC[60] (RRID:Addgene_101361) or ChrimsonR[61] (RRID: Addgene_59171) under control of the synapsin1 promoter. When ChR2 is expressed in CA3 pyramidal cells using AAV9, paired-pulse ratios are identical to electrophysiological stimulation, indicating physiological release probability[62]. For electrophysiological recordings, slices were placed in the recording chamber of the microscope and continuously perfused with ACSF (ACSF 4/4) saturated with 95% O$_2$ and 5% CO$_2$, containing (in mM): 119 NaCl, 26.2 NaHCO$_3$, 11 D-glucose, 1 NaH$_2$PO$_4$, 2.5 KCl, 4 CaCl$_2$, 4 MgCl$_2$, and 0.03 D-serine (pH 7.4, 308 mOsm kg$^{-1}$). Whole-cell recordings from CA1 pyramidal cells were made with a Multiclamp 700B amplifier (Molecular Devices) under the control of Ephus software written in MATLAB (MathWorks)[63]. Patch pipettes (borosilicate glass) were pulled to obtain tip resistances of 3–4 MΩ when filled with (in mM): 135 Cs-MeSO$_4$, 4 MgCl$_2$, 4 Na$_2$-ATP, 0.4 Na-GTP, 10 Na$_2$-phosphocreatine, 3 ascorbate, and 10 HEPES (pH 7.2, 295 mOsm kg$^{-1}$).

CA1 neurons were electroporated 3–7 days before experiments to express either mCerulean-LZ, ER-EGFP, and tdimer2 (control) or mCerulean-MyoV DN, ER-EGFP, and tdimer2. To optically induce LTP, we used an established protocol[21]. Under visual control, a fluorescent CA1 pyramidal cell was patched and voltage-clamped at −70 mV. Nonmuscle actin (5 μM, APHL99, Cytoskeleton Inc.) was included in the pipette to prevent wash-out of plasticity[64]. Presynaptic input was triggered by paired light pulses (2 ms duration, ISI 40 ms, 475 nm for ChR2, 625 nm for ChrimsonR) at 0.1 Hz through an optic fiber (0.4 mm) placed above CA3. After ~10 min of baseline recording (32 °C), light-induced EPSCs were paired ten times with 100 ms postsynaptic depolarizations to −15 mV. To investigate mGluR-dependent LTD, 50 μM APV (Tocris) was added to the ACSF to block NMDA receptors. The pipette solution contained K-gluconate instead of Cs-MeSO$_4$ to enable plasticity induction in current clamp (CC). Light-induced baseline EPSCs (~1 nA amplitude) were recorded in CA1 pyramidal cells at RT. LTD was induced by 900 paired light pulses at 1 Hz in CC. In all plasticity experiments, EPSC size was averaged every 60 s and effects were assessed 30 min after the end of the induction protocol. Only recordings in which the series resistance remained below 20 MΩ were included in the analysis. The outcome of plasticity experiments did not depend on the type of presynaptic channelrhodopsin (ChR2 ET-TC vs ChrimsonR) and results were therefore pooled.

**Immunohistochemistry and correlative analysis**. After following live ER dynamics in the 2P microscope for at least 1 h, slices were submerged in PBS containing 2% of PFA and sucrose and left overnight at 4 °C. They were washed three times with PBS, submerged in PBS containing 30% sucrose and kept at −80 °C. After thawing and refreezing once more, the slices were brought to RT for 30 min, washed twice with PBS and left shaking in PBS/TritonX-100 1% overnight at RT. After washing twice with PBS, slices were incubated with anti-synaptopodin antibody 1:1000 (S9442, Sigma) for 24 h shaking at 4 °C. The secondary antibody (Alexa Fluor 488 goat anti-rat 1:1000, Life Technologies) was used for slice incubation during 6 h at RT while shaking. Slices were finally mounted on glass slides with ProLong Gold Antifade Reagent (Life Technologies) and covered with a glass coverslip. Imaging was performed on a confocal microscope (Olympus FV1000) using 488 nm and 568 nm laser lines. Analysis was performed in Fiji. Briefly, segments previously imaged live were aligned with confocal images after immunostaining to identify corresponding spines (correlative analysis). Colocalization of red (native tdimer2) and green (anti-synaptopodin) fluorescence was assessed in confocal 3D datasets using axial line profiles (z-profiles) through the center of each spine (Supplementary Fig. 1). As stratum radiatum neuropil is densely packed with postsynaptic structures, synaptopodin clusters appeared inside as well as outside the transfected (red fluorescent) neuron. Spines with synaptopodin clusters above or below the spine head were scored as negative.

**Spine GluA2 content analysis**. A SEP-tagged version of GluA2[24] (gift from Roberto Malinow, RRID: Addgene_24001) was inserted under the synapsin promoter in a pCI backbone. Neurons in the CA1 region were electroporated 5 days prior to imaging to express tdimer2, SEP-GluA2 and mCerulean-LZ (ctrl group) or tdimer2, SEP-GluA2, and mCerulean-MyoV DN (DN group). Z-stacks of dendritic stretches were taken at 980 nm to measure spine volume and SEP-GluA2 fluorescence. Identical laser power was used to acquire data from both groups. An image of the apical dendrite and soma was also taken. To later assign dendritic stretches to each experimental group, presence of cytosolic cerulean was assessed at 810 nm wavelength. For spine GluA2 content and volume quantification, a macro in Fiji was used for semiautomated analysis. Briefly, z-stacks were median filtered (1 pixel) and background corrected (50 pixel rolling ball radius). On maximum intensity projections, circular ROIs (~0.5–1 μm diameter) were drawn on spine heads identified in the tdimer2 fluorescence channel and maximum fluorescence values of tdimer2 and SEP-GluA2 channels were extracted. The median-filtered image of the apical dendrite was used to fill the microscope PSF in order to normalize spine fluorescence and even out influence of protein expression level. Red to green crosstalk (5%) was corrected. GluA2 content was obtained dividing the SEP-GluA2 fluorescence at individual spines by the fluorescence at the apical dendrite close to the soma (GluA2$_{spine}$ = GluA2$_{spine\_fluo}$/GluA2$_{dend}$). Spine volume was in turn similarly calculated using tdimer2 fluorescence values (Volume$_{spine}$ = tdimer2$_{spine\_fluo}$/tdimer2$_{dend}$). Spine GluA2 concentration was obtained dividing GluA2 content by spine volume (GluA2$_{spine}$/Volume$_{spine}$).

**MyoVa overexpression**. To express full-length, brain-spliced isoform of mouse MYO5a tagged at its N-terminus to mEmerald, we isolated a fragment encoding the tagged myosin from pmEmerald-C1-brMyo5a, a plasmid that corresponds to pEGFP-C1-brMyo5a[65] carrying mEmerald[66] instead of EGFP. The myosin-containing fragment was inserted under the synapsin promoter in a pCI vector. To simultaneously label the ER, we replaced the EGFP fluorophore from pMH4-hsyn-ER-EGFP with DsRed2 (Clontech). For testing expression of MyoVa-mEmerald, CA1 cells were imaged 4 days after electroporation with either pMH4-hsyn-ER-DsRed2 and pCI-hsyn-MyoVa-mEmerald or pMH4-hsyn-ER-DsRed2, pCI-hsyn-MyoVa-mEmerald and MyoV DN. For quantification of ER+ spines, we imaged cells 4–12 days after electroporation with either pMH4-hsyn-ER-DsRed2, pCI-hsyn-MyoVa-mEmerald and mCerulean-LZ (ctrl group) or pMH4-hsyn-ER-DsRed2, pCI-hsyn-MyoVa-mEmerald, mCerulean-LZ, and MyoV DN (DN group). DNA concentration of each construct in electroporation solutions was 20 ng μl$^{-1}$.

**Reporting summary**. Further information on research design is available in the Nature Research Reporting Summary linked to this article.

## Data availability

Raw data and reagents are available from T.G.O. on reasonable request. Source Data are provided with this paper.

## Code availability

Custom code used for analysis has been deposited online (https://gin.g-node.org/Alberto_Perez-Alvarez/Spine_volume_ER_detection).

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

## Acknowledgements

We thank Iris Ohmert for excellent technical assistance and Christine E. Gee, Brenna Fearey, and Maria Andres-Alonso for critical reading of the manuscript. Ingke Braren of the UKE Vector Facility produced AAV vectors. Our work was supported by grants from the German Research Foundation (FOR 2419, Project #278170285; SFB 936, Project #178316478) to T.G.O. and W.W., an EMBO Long Term Fellowship (ALTF1172-2011) to A.P.-A., and a Marie Curie FP7 Integration Grant (PCIG11-GA-2012-321905) to W.W.

## Author contributions

A.P.-A. performed and designed experiments, analyzed data, and wrote the manuscript. S.Y. performed uncaging experiments, C.S. developed software for the laser scanning microscope, J.A.H. provided reagents, W.W. provided reagents and feedback, T.G.O. designed the study and wrote the manuscript. All authors read, commented on, and approved the manuscript.

## Funding

## Competing interests

The authors declare no competing interests.
