## [Peer Review File · Nature Communications]

Reviewers' comments:

Reviewer #1 (Remarks to the Author):

This is a short, novel study examining the dynamics of ER membrane in dendritic spines. The authors find that this movement is not random but is associated with periods of high synaptic activity. Furthermore, the movement into spines regulates synaptic plasticity. The experiments are well done. I have only a few comments.

The experiments are done under conditions that push the system to extremes. For example, the glutamate uncaging experiments are done in zero Mg, a state that increases the calcium influx to levels at least 5X higher than in normal ACSF (more than achieved with neuromodulators). Can they report the results of uncaging in solutions containing normal Mg? In previous experiments (Holbro et al., 2009) that show mGluR mediated, large amplitude calcium release following glutamate uncaging, apparently in normal ACSF.

Also, as the authors state, using ChR activation to activate synapses causes a higher and more reliable level of transmitter release than electrically activated spikes. The authors should discuss the level of confidence they have that their results apply in physiological conditions.

They say that synaptopodin was found in ~90% of spines that had a spine apparatus. How frequently was synaptopodin found in spines without a spine apparatus?

For the voltage clamp experiments (Fig. 3g) the authors should indicate how far the synapses were from the somatic pipette. Also, they should discuss the accuracy of their measurements considering that spines are difficult to clamp (cf. Beaulieu-Laroche and Harnett, 2018).

Scale bars are needed for Figs. 2a and 3a. Also, the red dot in Fig. 2a is probably too small. Their own estimate suggests that this spot should be at least 0.35x0.35 μm .

Reviewer #2 (Remarks to the Author):

This manuscript by Perez-Alvarez et al. provides evidence to support the idea that transient ER entry into dendritic spines is correlated with periods of an increased synaptic activity. The authors also provide data supporting the notion the transient ER entry prevents the strength of individual synapses from becoming saturated. The manuscript sheds a new insight into the role of ER in regulation of plasticity at the level of individual synapses.

One major concern is that interpretation of the dominant-negative MyoVa experiments seems a bit simplistic and would benefit from expanding discussion on how MyoVa cargos other than ER might be involved in the observed effect (They do mention BDNF). For example, MyoVa interacts with the postsynaptic scaffolding protein GKAP (Shin et al., 2012) and is involved in targeting of transmembrane proteins including GluA1 (Lewis et al., 2009). The authors point out that "the process of synaptic strengthening might take several days to fully develop" (Line 174-175) – so it seems likely that the impaired ER motility represents just one aspect of the MyoVa manipulation that results in synaptic scaling.

Other comments:

1. The Introduction is rather weak – the second paragraph is simply a restatement of the abstract. One might expect a more thorough treatment of the literature regarding ER in spines.
2. Figure 1 – it would be worthwhile to state that where the magenta of tdimer2 and ER-EGFP green overlap - white appears as "a light green" is really only visible where synpo is illustrated and might confuse readers.
3. Fig. 1b: Is the horizontal structure and axon? If so, please indicate and you might also remark

on stability of SER in the axon during this time, or comment on the mobility of SER in axons.

4. Fig. 1c: Confusing, in some rows there are black squares for most of the row, which is then interrupted by a white square – meaning stable SER? How can Stable SER appear in a spine that was hardly present, or disappeared? For example, there is a 'black' row near the middle of the chart with two white spots and another row a bit above it that is fully black (no spine) with just one white spot. The chart data should be double checked for accuracy.

5. Fig. 1g: IHC – This column is very confusing. A lot of the synpo labelling appears to be outside the tdimer2 labelling in all three pictures. More explanation is needed to interpret these images. In addition, it would be helpful if all spine images were oriented in the same direction, namely flipped side for side in the "No ER" row. Furthermore, in the 'Transient and No ER' rows there is 'synpo' labeling just next to the dendrite, ECS, or spine. Please explain.

6. Fig 1g and h: The authors state that Synpo indicates where stable Spine Apparatus is located. How then, do the authors interpret the presence of Synpo in spines with transient ER and also in neighboring spines that have no ER present? This point should be elaborated in the Results description.

7. Fig. 1j: are these distributions significantly different? What is the effect size?

8. Line 74: vague 'This' opening the sentence – what is 'This' referring back to?

9. Lines 75-77: Originally did all of the initial spines lack prior to stimulation?

10. Fig. 2A, 2E: Volume rows – are very confusing – is the frame shifting, or are multiple structures being added to the dendrite by the time the session reaches the 'end' at far right side. If not, then each frame should be exactly centered on the same spine that underwent glutamate uncaging.

11. Figure 2b: I do not trust the Size of spines measurements? Also, is the z-axis resolution of the imaging sufficient to capture a change in shape (especially for the large spines) that could change volume out of the plane of visualization into the depth. Given data from serial EM that shows large spines undergo the greatest changes in synaptic area (which correlates with change in volume and head diameter), this measure is perhaps the most problematic in the imaging data. Some discussion and discretion are warranted. In fact, if one looks at the 'volume row of the spine with uncaging, the shape appears to change, however, there is also a 'volume' shadow behind the spine in the first 6 panels that appears to merge with the red-dot spine by the last two panels. Is an adjacent spine emerging? How are these issues controlled?

12. Fig 2c: Is this figure based on spines with no ER at the time of stimulation? Also, based on this figure, uncaging resulted in structural LTP and ER insertion 53% of the time, not 80% as mentioned in Line 75.

13. Line 82, "At the single synapse level, we often observed spine swelling...": How often?

14. Lines 91-92: The time course presented is not beyond 36 minutes. Might be a bit of an overstatement to indicate 'no sign of lasting structural plasticity' – as the next phase of lasting potentiation (> 1hr) had not yet begun.

15. Figure 3a: The illustrated images appear to be biased – the Controls have 2 large spine visualized, whereas the DNMyoVa has only one large spine visualized... however, it is possible given figure 3b that this is the only spine with a stable ER – i.e. there are only 2 in the whole dataset, right? It appears that the density of spines in the tdimer image from the DNMyoVa has a higher density of small spines than Control – is that reflective of the overall data?

16. Line 111, DNMyoVa experiment: Does this manipulation affect spine density or the total amount of ER in the dendritic shaft? Also, did the authors perform rescue experiments (DNMyoVa + wild-type MyoVa)?

17. Fig 3e: How is ER entry affected by postsynaptic depolarization?

18. Lines 125-126 state 2x as many AMPAR in ER+ as ER- but then in Fig 3g: This figure is key to the whole argument, but it is difficult to understand. First, the absolute responses are impossible to compare as the y axes do not match. Second, aren't the ER+ neurons thought to have a larger response than the ER- neurons, hence are the titles in the two panels (ER+ and ER-) switched? Also, what precludes the conclusion that ER motility and visiting the spine is necessary for LTP – perhaps the AMPA receptors are not being properly activated or Phosphorylated in the absence of SER, and hence LTP is not produced. The rigidity of the interpretation is not warranted by the data. If the AMPA receptors were anchored away from the release sites, even just to the sides it

might also explain why there is high potency but no LTP. LTD may be produced through a different mechanism... The conclusions need a bit more thought and analysis.

19. Line 162: I like the idea that motors are moving the SER around dynamically and the data from Cerebellar P-cells in culture is compelling (Wagner et al., 2011). However, most P-cells have resident SER in adults but none of the parallel-fiber spines contain a spine apparatus, and as I recall from Frotscher's work, synaptopodin does not label those spines. Is it possible that only SA-negative SER is dynamic? The discussion would benefit from a bit more comment on the degree to which SER and SER-SA is dynamic in the Hippocampal cultures.

20. Line 192, "DNMyoVa-expressing neurons retain previously formed spine apparatus...": Was this confirmed by immunolabeling for synaptopodin, or simply based on ER being persistent/stable? (Fig 3a). Please See comment 20 also as it is relevant to this comment as well.

21. Line 273, "custom written software in Matlab": Please make sure to follow the journal's guideline for disclosing custom codes.

22. Line 288, "5 μ M non-muscle actin" (also Fig 3d): It would be useful if the authors could provide a rationale for this statement and use.

23. Line 339, Chirillo et al., 2015: This reference should be updated to Chirillo et al., 2019 and the effect of shaft SER on spine clustering around enlarged spines with SA or SER discussed in the context of the reported findings (doi: 10.1038/s41598-019-40520-x). Chirillo, M.A., et al., Local resources of polyribosomes and SER promote synapse enlargement and spine clustering after long-term potentiation in adult rat hippocampus. *Sci Rep*, 2019. 9(1): p. 3861.

References

Shin S. M., Zhang N., Hansen J., Gerges N. Z., Pak D. T., Sheng M., Lee S. H. (2012) GKAP orchestrates activity-dependent postsynaptic protein remodeling and homeostatic scaling. *Nat. Neurosci.* 15, 1655–1666

Lewis T. L. Jr., Mao T., Svoboda K., Arnold D. B. (2009) Myosin-dependent targeting of transmembrane proteins to neuronal dendrites. *Nat. Neurosci.* 12, 568–576

Chirillo, M.A., et al., Local resources of polyribosomes and SER promote synapse enlargement and spine clustering after long-term potentiation in adult rat hippocampus. *Sci Rep*, 2019. 9(1): p. 3861.

Reviewer #3 (Remarks to the Author):

The manuscript by Perez-Alvarez et al. documents transient entry of dendritic ER ("visits") into dendritic spines in hippocampal slice cultures that last for a few minutes. They show that such visits are non-random because they target spines during periods of high synaptic activity. Blockade of synaptic transmission in the organotypic cultures strongly reduces the number of short "visits." In spontaneously active cultures, visits are correlated with transient volume expansion of the spine, which is correlated with activity. They also show that transient ER entry into a spine is triggered by activation of the spine by glutamate uncaging. Transfection with a dominant-negative inhibitor of myosin-Va blocks the transient entry of ER into spines. This series of experiments is quite interesting.

The authors show that chronically blocking Myosin-Va function in the cultures (over 3 to 7 days) results in spines having considerably higher AMPA and NMDA-receptor currents than control spines upon whole cell stimulation. This result is puzzling because a strong earlier study (cited and discussed by the authors) showed that myosin-Va is necessary for transport of GluA1 receptors into the spine. Nevertheless, the authors point out that the earlier study examined neurons after ~15 hours of blockade; whereas their study examines neurons after 3 to 7 days of blockade. The authors postulate that a slow acting homeostatic process that depends upon transient entry of ER into spines prevents "additive" potentiation. Thus, blockade of ER entry into spines allows an unchecked build-up of AMPA and NMDA receptors. It appears that the build-up of AMPA receptor current must depend upon entry and immobilization of GluR2 into spines in the Myosin-Va DN cultures. The authors could support their hypothesis and clarify potential mechanisms by determining whether the AMPA current in these spines is carried predominantly by excess GluR2.

The authors discussion of potential mechanisms for a homeostatic role of the transient ER entry is weakened by two issues. The authors posit the potential involvement of IP3-dependent calcium-induced calcium release from the ER. However, the role of IP3 in LTD has been shown in Purkinje cells but is not as strongly supported in hippocampal neurons where the mechanism of LTD appears to be different. Second, blockade of myosin-VA in hippocampal pyramidal neurons would be expected to block transport of other components into the spine in addition to GluA1 and ER. For this reason, the direct involvement of ER entry in the mechanism by which blockade of myosin-VA causes an increase in AMPA and NMDA-receptors currents and blocks LTP is still uncertain.

There are two smaller points:

1. To validate the dominant-negative construct that the authors use to inhibit myosin-Va, the authors cite a paper that was uploaded to bioRxiv in January, 2019. Apparently this article has not yet been published after peer-review. The authors should cite a peer-reviewed paper to support the use of their construct. That said, the authors do show clearly that the DN construct blocks transient entry of ER into spines in their hippocampal cultures.

2. The authors do not indicate in their figures how many individual slice cultures or animals were used in each experiment. They only indicate the number of spines that were examined. For a complete understanding of the statistical analyses, it is important to state the number of slice cultures, and the number of animals from which the cultures were derived, for each experiment.

Response to reviewers' comments

Reviewer #1:

This is a short, novel study examining the dynamics of ER membrane in dendritic spines. The authors find that this movement is not random but is associated with periods of high synaptic activity. Furthermore, the movement into spines regulates synaptic plasticity. The experiments are well done. I have only a few comments.

The experiments are done under conditions that push the system to extremes. For example, the glutamate uncaging experiments are done in zero Mg, a state that increases the calcium influx to levels at least 5X higher than in normal ACSF (more than achieved with neuromodulators). Can they report the results of uncaging in solutions containing normal Mg? In previous experiments (Holbro et al., 2009) that show mGluR mediated, large amplitude calcium release following glutamate uncaging, apparently in normal ACSF.

To make it clear that we performed two very different kinds of uncaging experiments, we now split the section in the methods in two parts (lines 310-336). To assess AMPA and NMDAR currents on individual spines (Fig. 4c) we uncaged glutamate in high (3 mM) Mg²⁺, using postsynaptic depolarization to remove the Mg²⁺ block of NMDARs.

Glutamate uncaging in zero Mg²⁺ and TTX was used to induce structural LTP at single spines (Fig. 2a-d). The main advantage of this protocol is its high specificity and high reliability: structural LTP is induced at every spine stimulated this way. It has become the de-facto standard for single-synapse LTP studies (e.g. Kasai, Svoboda, Yasuoda, Malinow labs). We explain the rationale behind using the zero Mg²⁺ protocol in the discussion (lines 235-239):

“The resulting very strong NMDA currents might be able to override the homeostatic function of spine ER, leading to successful induction of LTP. This situation is mimicked by glutamate uncaging in zero Mg²⁺ saline, which also maximizes Ca²⁺ influx into the spine and provides an extremely strong stimulus. Under these conditions, we could induce lasting structural LTP in spite of ER entry. “

Also, as the authors state, using ChR activation to activate synapses causes a higher and more reliable level of transmitter release than electrically activated spikes. The authors should discuss the level of confidence they have that their results apply in physiological conditions.

The elevated release probability reported for the original ‘wild-type’ ChR2 (Zhang & Oertner 2007) is much less pronounced when using more recent, fast-closing variants (ChR2 ET-TC, ChrimsonR), which is why we used these variants in the present study. For transduction, we used AAV7 and AAV9 which do not cause changes in paired-pulse ratio (Jackman, Beneduce, Drew & Regehr, J Neurosci. 2014). We are quite confident that spike induction with light does not lead to fundamentally different forms of plasticity than spike induction with electrodes (c.f. Savic, Lüthi, Gähwiler & McKinney, PNAS 2003). In acute slices, the CA3 connections are usually severed to avoid epileptiform activity when inducing LTD/LTP. In organotypic cultures, using optogenetic activation of CA3 neurons, we can avoid recurrent activity without severing the connection between CA3 and CA1. Which situation is more physiological is difficult to guess.

They say that synaptopodin was found in ~90% of spines that had a spine apparatus. How frequently was synaptopodin found in spines without a spine apparatus?

To provide better quantification, we expanded the following section (l. 57-65):

“We hypothesized that these stably ER-positive spines contained a spine apparatus, and using 3D image stacks (Supplementary Fig. 1), we could indeed confirm synaptopodin immunoreactivity in 90% of these spines. In contrast, only 16% of spines with transient ER visits were scored as synaptopodin-positive (Fig. 1g, h). Within this group, spines that were scored ER-positive right before fixation were more likely to contain synaptopodin (20%, Supplementary Fig. 2). 4% of the spines that were never visited by ER stained positive against synaptopodin (Fig. 1g, h), which could indicate accumulation of this soluble protein prior to ER visits.”

For the voltage clamp experiments (Fig. 3g) the authors should indicate how far the synapses were from the somatic pipette. Also, they should discuss the accuracy of their measurements considering that spines are difficult to clamp (cf. Beaulieu-Laroche and Harnett, 2018).

We added the following text to the methods section (line 329-336):

“For both control and MyoV DN groups, uncaged spines were located on thin oblique dendrites at a distance equal or below 100 μm from the somatic recording pipette. Due to the high electrical resistance of spine necks, voltage-clamp of spine synapses is far from perfect (Beaulieu-Laroche & Harnett, 2018; Grunditz, Holbro, Tian, Zuo, & Oertner, 2008). Thus, the true synaptic conductance may be higher than our measured EPSCs suggest. However, as the diffusional coupling of ER-negative and ER-positive spines is similar (Holbro, Grunditz, & Oertner, 2009), we have no reason to suspect systematic differences in spine neck resistance between control and MyoV DN groups. Thus, the large difference in EPSC amplitude most likely reflects a proportional increase in the number of glutamate receptors per spine in MyoV DN-expressing neurons.”

Scale bars are needed for Figs. 2a and 3a. Also, the red dot in Fig. 2a is probably too small. Their own estimate suggests that this spot should be at least 0.35x0.35 μm .

Scale bars have been added and the dot size, which was supposed to be symbolic, has been corrected.

Reviewer #2:

This manuscript by Perez-Alvarez et al. provides evidence to support the idea that transient ER entry into dendritic spines is correlated with periods of an increased synaptic activity. The authors also provide data supporting the notion the transient ER entry prevents the strength of individual synapses from becoming saturated. The manuscript sheds a new insight into the role of ER in regulation of plasticity at the level of individual synapses.

One major concern is that interpretation of the dominant-negative MyoVa experiments seems a bit simplistic and would benefit from expanding discussion on how MyoVa cargos other than ER might be involved in the observed effect (They do mention BDNF). For example, MyoVa interacts with the postsynaptic scaffolding protein GKAP (Shin et al., 2012) and is involved in targeting of transmembrane proteins including GluA1 (Lewis et al., 2009). The authors point out that the process of synaptic strengthening might take several days to fully develop (Line 174-175); so it seems likely that the impaired ER motility represents just one aspect of the MyoVa manipulation that results in synaptic scaling.

We agree with the reviewer that other cargos might be affected by impairing endogenous MyoVa function. In fact, the dominant negative approach might also affect cargo binding of MyoVb, and we changed the shorthand to MyoV DN to indicate this fact (consistent with Konietzny et al. 2020). We have expanded the discussion regarding the topic of specificity and other cargos (line 183-206):

“When comparing results of dominant-negative interference with myosin V on synaptic strength and plasticity, it is important to point out that myosin Va has cargo binding sites on both its globular tail

domain (GTD) and its stalk (coiled-coil region). Our DN construct with leucine zipper dimerization was designed to compete with intact myosin Va specifically for its GTD-binding cargo since it lacks the myosin Va coiled-coil (Konietzny et al. 2019). As we expected, expression of myoV DN blocked ER motility almost completely, but it may also have affected the distribution of other GTD-binding cargos. Of obvious importance for synaptic plasticity is the GluA1 subunit of the AMPA receptor (Correia et al. 2008), which may be transported by MyoVa or MyoVb isoforms (Lewis Jr. et al. 2009; Wang et al. 2008). Interestingly, hippocampal LTP is intact in the MyoVa null mutant dilute-lethal, suggesting that both MyoV isoforms may be able to traffic GluA1 (Schnell and Nicoll 2001). While disturbed GluA1 delivery could explain the lack of LTP we observed after MyoV DN expression, it does not explain why both AMPA and NMDA responses grew stronger and spine surface expression of GluA2 was enhanced.

Exocytosis and transport of dense-core vesicles is increased by a MyoVa DN (Bittins, Eichler, and Gerdes 2009), possibly enhancing release of BDNF, an important regulator of synaptic strength. BDNF release from spines is necessary for LTP induction (Harward et al. 2016) and might contribute to the increased baseline synaptic strength in MyoV DN neurons. However, BDNF release from dendrites and spines is unlikely to be from dense-core vesicles and the mechanism has not been determined.

There are other MyoVa binding proteins affecting synaptic plasticity such as GKAP, a scaffold protein accumulating in spines in an inverse relation to activity and CaMKII α (Ehlers 2003). Overexpression of a non-functional GKAP mutant impaired homeostatic synaptic scaling in both directions, but had no effect on the frequency or amplitude of spontaneous synaptic currents (Shin et al. 2012). GKAP is thought to bind to the myosin stalk (Naisbitt et al. 2000), so we would not expect it to be affected by our DN construct.

While we acknowledge the limited specificity of the DN approach (Lewis et al. 2009) and the naiveté of monocausal explanations, it is difficult to imagine how disruption of cargo transport or anchoring would lead to an upwards drift of synaptic strength unless the cargo is a negative regulator of synaptic strength.”

Other comments:

1. The Introduction is rather weak - the second paragraph is simply a restatement of the abstract. One might expect a more thorough treatment of the literature regarding ER in spines.

We expanded the introduction to cover seminal ER literature (p. 2).

2. Figure 1 - it would be worthwhile to state that where the magenta of tdimer2 and ER-EGFP green overlap - white appears as “a light green” is really only visible where synpo is illustrated and might confuse readers.

We removed the green lettering (ER-EGFP) from the figure, as it might confuse readers. In the figure legend, we state now that the green-labeled ER appears white when printed over magenta. (This is a disadvantage of the color-blind friendly palette.)

3. Fig. 1b: Is the horizontal structure and axon? If so, please indicate and you might also remark on stability of SER in the axon during this time, or comment on the mobility of SER in axons.

Indeed, the horizontal structure in Fig. 1b is an axon, which we now explain in the legends of Fig 1 and Supplementary Video 1. Being out of scope of this study, we did not quantify the dynamics of axonal ER. We typically observed a dim but stable ER signal, consistent with the reported presence of thin ER tubes in axons (Terasaki et al., 2018).

4. Fig. 1c: Confusing, in some rows there are black squares for most of the row, which is then interrupted by a white square - meaning stable SER? How can Stable SER appear in a spine that was hardly present, or disappeared? For example, there is a 'black'; row near the middle of the chart with two white spots and another row a bit above it that is fully black (no spine) with just one white spot. The chart data should be double checked for accuracy.

The score chart is now better explained in the methods section (l. 294-300), figure legend, and by adding a legend of categories above the chart.

“In score charts of time-lapse experiments, white color denotes GFP-signal from the ER present inside a tdimer2-filled spine (green over magenta adds to white). At time points when a spine could not be resolved (i.e. retraction or proximity $<0.5 \mu\text{m}$ to the dendrite axis), no ER scoring could be allocated (black color). If a spine appeared intermittently in the same location, it was scored in the same chart row. If ER entered at least once the dendritic spine head or neck, spines were scored as “Transient ER”. If ER was present in the spine throughout the experiment, it was assigned to the “Stable ER” group. Spines that never experienced ER entry were classified as “No ER”.”

5. Fig. 1g: IHC - This column is very confusing. A lot of the synpo labelling appears to be outside the tdimer2 labelling in all three pictures. More explanation is needed to interpret these images. In addition, it would be helpful if all spine images were oriented in the same direction, namely flipped side for side in the 'No ER' row. Furthermore, in the 'Transient and No ER' rows there is 'synpo' labeling just next to the dendrite, ECS, or spine. Please explain.

All our experiments were performed on hippocampal tissue, not on dissociated neurons. While cytoplasmic and ER label is restricted to individually transfected neurons (and therefore is very sparse), anti-synaptopodin immunostaining visualizes the protein in ALL cells, therefore appearing inside as well as outside transfected neurons. We expanded our explanation in the legend of Fig. 1:

“g, Three examples (two-photon time series, maximum intensity projections) of spines with stable (top), transient (middle) or no ER (bottom), followed by correlative confocal images (maximum intensity projections) of the same spines (native tdimer2 fluorescence, red) after fixation of the tissue and immunostaining against synaptopodin (cyan). Scale bars: $1 \mu\text{m}$. Note synaptopodin clusters inside (white) as well as outside (cyan) the transfected neuron (red), as the antibody labels synaptopodin in the entire neuropil.”

To better explain that our scoring was not based on the two-dimensional projection shown in the figures, but involved analysis in 3D, we added a supplemental figure (S1) and appended the methods section (l. 374-381):

“Briefly, segments previously imaged live were aligned with confocal images after immunostaining to identify corresponding spines (correlative analysis). Colocalization of red (native tdimer2) and green (anti-synaptopodin) fluorescence was assessed in confocal 3D datasets using axial line profiles (z-profiles) through the center of each spine (Supplementary Figure 1). As stratum radiatum neuropil is densely packed with postsynaptic structures, synaptopodin clusters appeared inside as well as outside the transfected (red fluorescent) neuron. Spines with synaptopodin clusters above or below the spine head were scored as negative.”

We flipped the images of the 'no ER' row as requested.

6. Fig 1g and h: The authors state that Synpo indicates where stable Spine Apparatus is located. How

then, do the authors interpret the presence of Synpo in spines with transient ER and also in neighboring spines that have no ER present? This point should be elaborated in the Results description.

We appended the Results (l. 57-64):

“We hypothesized that these stably ER-positive spines contained a spine apparatus, and using 3D image stacks (Supplementary Fig. 1), we could indeed confirm synaptopodin immunoreactivity in 90% of these spines. In contrast, only 16% of spines with transient ER visits were scored as synaptopodin-positive (Fig. 1g, h). Within this group, spines that were scored ER-positive right before fixation were more likely to contain synaptopodin (20%, Supplementary Fig. 2). 4% of the spines that were never visited by ER stained positive against synaptopodin (Fig. 1g, h), which could indicate accumulation of this soluble protein prior to ER visits (Konietzny et al. 2020).”

7. Fig. 1j: are these distributions significantly different? What is the effect size?

Yes, the two distributions are different (Mann Whitney test, $p = 0.0005$). This is included in the text. The effect size is difficult to assess as most spines were never visited by ER. Thus, both distributions are far from normal, they are extremely skewed. ‘Mean ER residence time’ is therefore not a very informative measure (Cohen’s $d = 0.1$).

8. Line 74: vague ‘This’ opening the sentence - what is ‘This’ referring back to?

We apologize for the lack of clarity. The text has now been rewritten (l. 76-78):

“The uncaging protocol induced a lasting volume increase of spines that had no ER at baseline, but had no consistent effect on ER+ spines (Fig. 2a, b). This difference in structural plasticity could be due to efficient removal of free Ca^{2+} by SERCA pumps (Verkhratsky, 2005) in ER+ spines.”

9. Lines 75-77: Originally did all of the initial spines lack prior to stimulation?

Spines were scored ER+ or ER- based on presence or absence of ER, respectively, right before uncaging. Indeed, all ER-negative spines lacked ER before stimulation.

10. Fig. 2A, 2E: Volume rows - are very confusing - is the frame shifting, or are multiple structures being added to the dendrite by the time the session reaches the ‘end’ at far right side. If not, then each frame should be exactly centered on the same spine that underwent glutamate uncaging.

We improved the alignment of the images. The dynamics of the neighboring spines is real, however, and we don’t see a reason to hide this by restricting the frame to the stimulated spine only.

11. Figure 2b: I do not trust the Size of spines measurements? Also, is the z-axis resolution of the imaging sufficient to capture a change in shape (especially for the large spines) that could change volume out of the plane of visualization into the depth. Given data from serial EM that shows large spines undergo the greatest changes in synaptic area (which correlates with change in volume and head diameter), this measure is perhaps the most problematic in the imaging data. Some discussion and discretion are warranted. In fact, if one looks at the -volume row of the spine with uncaging, the shape appears to change, however, there is also a ‘volume’ shadow behind the spine in the first 6 panels that appears to merge with the red-dot spine by the last two panels. Is an adjacent spine emerging? How are these issues controlled?

This commentary, mentioning the danger that the spine may go partially out of focus, indicates a misunderstanding about our imaging strategy. At every time point, a 3D volume much larger than the spine was sampled. In the figures, we are obviously forced to show 2D projections, but all our analysis was done on the full 3D datasets. We added Supplementary Figure 1 to illustrate this fact. Only spine heads that were clearly distinguishable from other structures in all directions (x,y,z) were included in the analysis. To clarify how we measured spine volume changes upon glutamate uncaging, we added the following paragraph to the methods section (l. 317-322):

“Morphological analysis was performed using custom written software (MATLAB, MathWorks). For a relative measure of spine volume, spine fluorescence was normalized to local dendritic fluorescence. Briefly, a circular ROI was centered on the spine head at the time point where the uncaged spine reached its largest volume. Within the ROI, the brightest 20% of pixels were averaged in every z-plane. The maximum fluorescence, corresponding to the z-plane intersecting the center of the spine, was selected and normalized to the local dendritic fluorescence.”

12. Fig 2c: Is this figure based on spines with no ER at the time of stimulation? Also, based on this figure, uncaging resulted in structural LTP and ER insertion 53% of the time, not 80% as mentioned in Line 75.

Fig. 2c indeed shows a breakdown of spines with no ER at the time of stimulation. We now added small icons to make this more clear. For quantification, we considered those spines that successfully underwent structural changes upon uncaging (10/15). In Fig.2b and 2c we also show how many spines did not experience any change (5/10).

The text (l. 78) has been rewritten for clarity:

“In 8 out of 10 spines showing structural LTP, ER was immediately attracted into the stimulated spine (Fig. 2c, d; Supplementary Movie 2)...”

13. Line 82, “At the single synapse level, we often observed spine swelling”: How often?

To avoid unspecific terms, we have rewritten the section (l. 86-92):

“We observed spine enlargement of variable intensity and duration preceding the ER visit, followed by rapid collapse back to baseline volume after ER retraction (Fig. 2e). As ER visits occurred at different time points, the average fluorescence intensity across all monitored synapses was constant over the course of the experiment (Fig. 2f, left). When we aligned the spine head volume traces (tdimer2 fluorescence intensity normalized to the local dendrite) of all recorded spines to the time point when they reached their peak volume, we found that at the time the spine head was largest ($t = 0$), the probability of ER invasion was maximal (Fig. 2f, center).”

14. Lines 91-92: The time course presented is not beyond 36 minutes. Might be a bit of an overstatement to indicate ‘no sign of lasting structural plasticity’ - as the next phase of lasting potentiation (> 1hr) had not yet begun.

Our statement refers not to the single example showing only one isolated ER insertion event (panel 2e), but to the aggregated data (panel 2f), where spine volume and ER visits was tracked for several hours. We found no statistical difference in the average spine volume data in the 200 min before and the 200 min after the ER visit. Therefore, our statement is correct.

15. Figure 3a: The illustrated images appear to be biased - the Controls have 2 large spine visualized, whereas the DNMyoVa has only one large spine visualized - however, it is possible given figure 3b that

this is the only spine with a stable ER - i.e. there are only 2 in the whole dataset, right? It appears that the density of spines in the t-dimer image from the DNMyoVa has a higher density of small spines that Control - is that reflective of the overall data?

We have trouble understanding this accusation of bias. The Control dataset has 4 spines with stable ER and we show 2, the MyoVa DN dataset has 2 spines with stable ER and we show 1. Therefore, the proportion of stable ER spines is exactly representative in the displayed stretches of dendrite.

Spine density is slightly different in the chosen example, but this was not the case when we compared all analyzed dendrites. To clarify this point, we added a plot of spine densities in control and MyoVa DN dendrites (Figure 3d).

16. Line 111, DNMyoVa experiment: Does this manipulation affect spine density or the total amount of ER in the dendritic shaft? Also, did the authors perform rescue experiments (DNMyoVa + wild-type MyoVa)?

We show now quantification of spine density in Fig. 3d. We did not observe any obvious differences in dendritic ER (see new Supplementary Fig. 4), but light microscopy cannot resolve the intricate morphology of dendritic ER networks. We now cite EM studies that investigated dendritic ER in detail (Spacek & Harris 1997; Chirillo et al. 2019).

In response to the reviewer's suggestion, we performed MyoVa overexpression experiments (new Supplementary Figure 5). Indeed, MyoVa overexpression increased the number of ER-containing spines compared to control neurons, suggesting that the availability of motors is a limiting factor for ER motility. When co-expressed with the DN construct, we observed partial rescue, but not to the full level of control neurons (expressing only fluorescent labels). This outcome fits with the proposed mechanism of action of the DN construct, competing with functional MyoVa (both endogenous and overexpressed) for ER binding.

17. Fig 3e: How is ER entry affected by postsynaptic depolarization?

We did not monitor ER motility during optogenetic LTP induction (the bright blue light pulses interfere with simultaneous optical measurements). The effect of postsynaptic depolarization was investigated in uncaging-induced LTP experiments (Fig. 2a-d): ER is drawn into depolarized spines.

18. Lines 125-126 state 2x as many AMPAR in ER+ as ER- but then in Fig 3g: This figure is key to the whole argument, but it is difficult to understand. First, the absolute responses are impossible to compare as the y axes do not match. Second, aren't the ER+ neurons thought to have a larger response than the ER- neurons, hence are the titles in the two panels (ER+ and ER-) switched? Also, what precludes the conclusion that ER motility and visiting the spine is necessary for LTP - perhaps the AMPA receptors are not being properly activated or Phosphorylated in the absence of SER, and hence LTP is not produced. The rigidity of the interpretation is not warranted by the data. If the AMPA receptors were anchored away from the release sites, even just to the sides it might also explain why there is high potency but no LTP. LTD may be produced through a different mechanism... The conclusions need a bit more thought and analysis.

We profusely apologize for the misplacement of the panel titles. This has now been fixed, and axes of both panels match (now Fig. 4c).

To strengthen our argument that spines contain more AMPA receptors after prolonged block of MyoVa, we performed additional experiments, using pH-sensitive AMPA receptor labeling (SEP-GluA2) to measure surface expression on spines (new Supplementary Fig. 6). Indeed, spines on MyoV DN dendrites

have significantly more AMPA receptors (new Fig. 4a, b). We hope this new data convinces the reviewer that our original interpretation (that MyoV DN leads to strengthened synapses) was correct.

We don't think ER visits to spines are required for LTP. We have reason to believe that the DN construct used by Correia et al. did not block ER visits (now explained in line 178), and yet they report complete block of LTP. We use a dimerizing DN construct (which does block ER visits) and also find no LTP.

The reviewer correctly points out that our methods do not distinguish extra- and intrasynaptic receptors, and we spell this out in the revised discussion (l. 176-182):

"A pioneering study expressing a different MyoVa-GTD construct in CA1 neurons also reported GluA2 enrichment in spines and blocked LTP, but in contrast to our results, AMPAR currents were reduced 15 hours after transduction (Correia et al., 2008). Presence of spine ER was not assessed by Correia et al., but in our hands, a non-dimerizing GTD as they used does not affect ER spine targeting (Konietzny et al. 2019). Thus, it is possible that the potentiation of baseline strength we report here happens only when spine synapses are not visited by ER for several days, or it reflects an increase in extrasynaptic receptors."

19. Line 162: I like the idea that motors are moving the SER around dynamically and the data from Cerebellar P-cells in culture is compelling (Wagner et al., 2011). However, most P-cells have resident SER in adults but none of the parallel-fiber spines contain a spine apparatus, and as I recall from Frotscher's work, synaptopodin does not label those spines. Is it possible that only SA-negative SER is dynamic? The discussion would benefit from a bit more comment on the degree to which SER and SER-SA is dynamic in the Hippocampal cultures.

Indeed, we believe that spines with dynamic ER do not contain a spine apparatus, and we provide data to support this idea (l. 57-62):

"We hypothesized that these stably ER-positive spines contained a spine apparatus, and using 3D image stacks (Supplementary Fig. 1), we could indeed confirm synaptopodin immunoreactivity in 90% of these spines. In contrast, only 16% of spines with transient ER visits were scored as synaptopodin-positive (Fig. 1g, h). Within this group, spines that were scored ER-positive right before fixation were more likely to contain synaptopodin (20%, Supplementary Fig. 2)."

We further comment in the discussion (l. 161-170):

"In our study, spines with transient ER visits were mostly synaptopodin-negative. This is not surprising, as the process of spine apparatus assembly inside a spine takes about 1 hour (Konietzny 2019). Indeed, we detected synaptopodin in 90% of spines with stable ER, and after several days of MyoV DN expression, the fraction of these stable ER spines was halved. This suggests that once a spine apparatus is assembled and sufficiently stabilized by synaptopodin, myosin activity is not constantly required for its maintenance (Konietzny 2019). The situation might be different in Purkinje cells, which do not express synaptopodin, do not form spine apparatus (Deller 2003), but still are able to sustain tubular ER in every spine (Wagner 2011). Structurally, Purkinje cell spines are much less dynamic than CA1 pyramidal cell spines and regulate actin organization in a distinct manner (Roesler 2019). Therefore, organization of the actin tracks, not the processive motor protein itself, may be responsible for the differences in spine ER stability."

20. Line 192, "DNMyoVa-expressing neurons retain previously formed spine apparatus": Was this confirmed by immunolabeling for synaptopodin, or simply based on ER being persistent/stable? (Fig 3a). Please See comment 20 also as it is relevant to this comment as well.

Indeed we did not confirm the existence of a true spine apparatus by immunostaining in these experiments. Therefore, to be more precise, we rephrased the section (l. 104-110):

“Expression of MyoV DN in CA1 neurons decreased five-fold the proportion of spines that contained ER, from 20% to 4% (Fig. 3a-c, Supplementary Fig. 4) while the density of spines on the dendrite was similar in both groups (Fig. 3d). Transient ER visits were extremely rare in MyoV DN neurons, decreasing from 41% to 3% (Fig. 3e, Supplementary Video 3). A small number of spines (2%) still had stable ER in MyoV DN-expressing neurons, suggesting that spines with already stable presence of ER, likely containing a spine apparatus, did not require myosin Va for its maintenance (Fig. 3e, Supplementary Fig. 4).”

21. Line 273, “custom written software in Matlab”: Please make sure to follow the journal’s guideline for disclosing custom codes.

Link to custom code repository is now provided.

22. Line 288, “5 μ M non-muscle actin” (also Fig 3d): It would be useful if the authors could provide a rationale for this statement and use.

We added the following explanation (l. 352): “Non-muscle actin (5 μ M, APHL99, Cytoskeleton Inc.) was included in the pipette to prevent wash-out of plasticity (Tanaka et al., 2008).”

23. Line 339, Chirillo et al., 2015: This reference should be updated to Chirillo et al., 2019 and the effect of shaft SER on spine clustering around enlarged spines with SA or SER discussed in the context of the reported findings (doi: 10.1038/s41598-019-40520-x). Chirillo, M.A., et al., Local resources of polyribosomes and SER promote synapse enlargement and spine clustering after long-term potentiation in adult rat hippocampus. Sci Rep, 2019. 9(1): p. 3861.

This reference has now been updated and commented in the discussion (l. 154-158):

“When we used 2P glutamate uncaging to induce structural LTP at single spines, we transiently attracted ER into the swelling spines and did not observe stabilization of spine ER that would suggest formation of a spine apparatus. Possibly multiple synapses need to simultaneously be potentiated (as with theta-burst stimulation) for the redistribution of dendritic ER towards potentiated spines, which may be necessary to provide material for spine apparatus formation (Chirillo et al., 2019).”

Reviewer #3 (Remarks to the Author):

The manuscript by Perez-Alvarez et al. documents transient entry of dendritic ER (‘visits’) into dendritic spines in hippocampal slice cultures that last for a few minutes. They show that such visits are non-random because they target spines during periods of high synaptic activity. Blockade of synaptic transmission in the organotypic cultures strongly reduces the number of short ‘visits’. In spontaneously active cultures, visits are correlated with transient volume expansion of the spine, which is correlated with activity. They also show that transient ER entry into a spine is triggered by activation of the spine by glutamate uncaging. Transfection with a dominant-negative inhibitor of myosin-Va blocks the transient entry of ER into spines. This series of experiments is quite interesting. The authors show that chronically blocking Myosin-Va function in the cultures (over 3 to 7 days) results in spines having considerably higher AMPA and NMDA-receptor currents than control spines upon whole cell stimulation. This result is puzzling because a strong earlier study (cited and discussed by the authors) showed that myosin-Va is necessary for transport of GluA1 receptors into the spine. Nevertheless, the authors point out that the earlier study examined neurons after ~15 hours of blockade; whereas their

study examines neurons after 3 to 7 days of blockade. The authors postulate that a slow acting homeostatic process that depends upon transient entry of ER into spines prevents “additive” potentiation. Thus, blockade of ER entry into spines allows an unchecked build-up of AMPA and NMDA receptors. It appears that the build-up of AMPA receptor current must depend upon entry and immobilization of GluR2 into spines in the Myosin-Va DN cultures. The authors could support their hypothesis and clarify potential mechanisms by determining whether the AMPA current in these spines is carried predominantly by excess GluR2.

In response to this suggestion, we performed additional experiments to directly visualize GluR2-containing receptors at spines (SEP-GluA2). The results, presented in the new Fig. 4 and new Supplemental Fig. 5, confirm that MyoVa DN-expressing neurons indeed have excess GluR2 on spines.

The authors discussion of potential mechanisms for a homeostatic role of the transient ER entry is weakened by two issues. The authors posit the potential involvement of IP3-dependent calcium-induced calcium release from the ER. However, the role of IP3 in LTD has been shown in Purkinje cells but is not as strongly supported in hippocampal neurons where the mechanism of LTD appears to be different.

Please check the seminal paper from Oliet, Malenka & Nicoll (1997), entitled “Two distinct forms of long-term depression coexist in CA1 hippocampal pyramidal cells”. mGluR-dependent LTD in the hippocampus was firmly established by this paper and later confirmed by other labs (e.g. Snyder, Philpot, Huber & Bear, 2001). We have previously shown that under conditions of blocked NMDARs, Schaffer collateral LTD can still be induced by 1 Hz stimulation, but is only expressed at spines that contain ER (Holbro et al. 2009). In Holbro et al., we show block of mGluR-LTD by heparin, an inhibitor of the IP3 receptor.

Second, blockade of myosin-VA in hippocampal pyramidal neurons would be expected to block transport of other components into the spine in addition to GluA1 and ER. For this reason, the direct involvement of ER entry in the mechanism by which blockade of myosin-VA causes an increase in AMPA and NMDA-receptors currents and blocks LTP is still uncertain.

We have included new data (SEP-GluA2 imaging, rescue experiments) and further characterized the effects of MyoVa on spine density and volume. We propose a link between the lack of ER motility, growth of synapses and block of LTP as a possible mechanism that seems both plausible and parsimonious to us. Other explanations are certainly possible, and we substantially extended the manuscript to better explain advantages and limitations of the dominant-negative approach (l. 183-212):

“When comparing results of dominant-negative interference with myosin V on synaptic strength and plasticity, it is important to point out that myosin Va has cargo binding sites on both its globular tail domain (GTD) and its stalk (coiled-coil region). Our DN construct with leucine zipper dimerization was designed to compete with intact myosin Va specifically for its GTD-binding cargo since it lacks the myosin Va coiled-coil. As we expected, expression of myoV DN blocked ER motility almost completely, but it may also have affected the distribution of other GTD-binding cargos. Of obvious importance for synaptic plasticity is the GluA1 subunit of the AMPA receptor, which may be transported by MyoVa or MyoVb isoforms. Interestingly, hippocampal LTP is intact in the MyoVa null mutant dilute-lethal, suggesting that both MyoV isoforms may be able to traffic GluA1. While disturbed GluA1 delivery could explain the lack of LTP we observed after MyoV DN expression, it does not explain why both AMPA and NMDA responses grew stronger and spine surface expression of GluA2 was enhanced.

Exocytosis and transport of dense-core vesicles is increased by a MyoVa DN37, possibly enhancing release of BDNF, an important regulator of synaptic strength. BDNF release from spines is necessary for LTP induction and might contribute to the increased baseline synaptic strength in MyoV DN neurons. However, BDNF release from dendrites and spines is unlikely to be from dense-core vesicles and the mechanism has not been determined.

There are other MyoVa binding proteins affecting synaptic plasticity such as GKAP, a scaffold protein accumulating in spines in an inverse relation to activity and CaMKII α . Overexpression of a non-functional GKAP mutant impaired homeostatic synaptic scaling in both directions, but had no effect on the frequency or amplitude of spontaneous synaptic currents. GKAP is thought to bind to the myosin stalk, so we would not expect it to be affected by our DN construct.

While we acknowledge the limited specificity of the DN approach and the naiveté of monocausal explanations, it is difficult to imagine how disruption of cargo transport or anchoring would lead to an upwards drift of synaptic strength unless the cargo is a negative regulator of synaptic strength. Previously, we observed that only ER containing spines are able to undergo mGluR-dependent LTD. At that time, we were surprised how few hippocampal spines were competent to express LTD. Now, taking ER dynamics into account, we observe that within a 5 h window, most spines are visited by ER at least once and that these transient ER visits coincide with LTP-like events. In combination with the MyoV DN observations, we propose that in hippocampal neurons, ER visits strongly activated spines and limits potentiation by a group1 mGluR dependent mechanism.”

There are two smaller points:

1. To validate the dominant-negative construct that the authors use to inhibit myosin-Va, the authors cite a paper that was uploaded to bioRxiv in January, 2019. Apparently, this article has not yet been published after peer-review. The authors should cite a peer-reviewed paper to support the use of their construct. That said, the authors do show clearly that the DN construct blocks transient entry of ER into spines in their hippocampal cultures.

In the meantime, the preprint in question (Konietzny et al.) has been published in J. Cell Sci.; the correct reference is now cited. As Konietzny et al. contains a thorough characterization of the exact DN construct used in the present study, we also adopt its nomenclature (MyoV DN).

2. The authors do not indicate in their figures how many individual slice cultures or animals were used in each experiment. They only indicate the number of spines that were examined. For a complete understanding of the statistical analyses, it is important to state the number of slice cultures, and the number of animals from which the cultures were derived, for each experiment.

The numbers of cells and slice cultures have now been added in each of the figure legends.

REVIEWERS' COMMENTS:

Reviewer #1 (Remarks to the Author):

This revision answers the questions I raised in the original review although all issues are not settled cleanly.

It is true that the method to induce structural LTP in these experiments (uncaging in the presence of zero Mg) is similar to that used by other excellent investigators. But that does not make it physiological. The increase in calcium with this method (compared to uncaging in normal Mg) is several times higher than the amount affected by neuromodulators in e.g. Giessel and Sabatini (cited by the authors). Some caution about this point is warranted.

What is the evidence or reference for the statement, "The elevated release probability reported for the original 'wild-type' ChR2 (Zhang & Oertner 2007) is much less pronounced when using more recent, fast-closing variants (ChR2 ET-TC, ChrimsonR)" (in the rebuttal).

Reviewer #2 (Remarks to the Author):

Our primary concerns have been addressed. We have a couple comments that would improve the clarity or provide consistency in style including the following:

(1) The use of the term "invade" when describing ER entry into spines gives the impression that ER is not supposed to go in there. We much prefer the more neutral description e.g., "entry" or "visits" (which the authors already use in the manuscript) for this description. We like "visits" because it fits the activity- and ATP-dependent nature of ER trafficking into spines by Myo V.

(2) The author could improve the transition to the last paragraph of Results section by mentioning why they are looking at surface expression of AMPAR. Although it seems obvious, the current transition is a bit abrupt.

Finally, we were unable to find in the manuscript the author's response to reviewer #1's comment two -- regarding ChR-mediated activation of synapses.

Reviewer #3 (Remarks to the Author):

The authors have responded thoroughly to the comments of all previous reviewers, and have added clarifying new material to the manuscript. The manuscript is now ready for publication.

Reviewer #1 (Remarks to the Author):

This revision answers the questions I raised in the original review although all issues are not settled cleanly.

It is true that the method to induce structural LTP in these experiments (uncaging in the presence of zero Mg) is similar to that used by other excellent investigators. But that does not make it physiological. The increase in calcium with this method (compared to uncaging in normal Mg) is several times higher than the amount affected by neuromodulators in e.g. Giessel and Sabatini (cited by the authors). Some caution about this point is warranted.

We now clarify this point in the Results section:

“This approach circumvents the presynaptic terminal and leads to maximal activation of postsynaptic NMDA receptors (Matzusaki et al., 2004).”

What is the evidence or reference for the statement, “The elevated release probability reported for the original ‘wild-type’ ChR2 (Zhang & Oertner 2007) is much less pronounced when using more recent, fast-closing variants (ChR2 ET-TC, ChrimsonR)” (in the rebuttal).

The statement reflects our own measurements. We have used ChrimsonR in CA3 neurons to induce STDP. Results were identical to electrically induced STDP under direct recording conditions (<https://www.biorxiv.org/content/10.1101/863365v3.full>). ChrimsonR closes with a time constant of ~8 ms at room temperature, which is approximately twice as fast as WT ChR2. (Sabatier et al.: Modeling the Electro-chemical Properties of Microbial Opsin ChrimsonR for Application to Optogenetics-based Vision Restoration. doi: <https://doi.org/10.1101/417899>.)

However, we would prefer not to include a lengthy discussion of channelrhodopsin kinetics in the current MS. Importantly, we do not use optogenetic stimulation to draw any conclusions about release probability. We show differential, genotype-specific effects on LTP and LTD. Thus, even if the release probability was higher with optogenetic stimulation than with traditional electrophysiological stimulation (which we claim is not the case), it would not change the conclusion that Myo Va overexpression changes long-term plasticity.

Reviewer #2 (Remarks to the Author):

Our primary concerns have been addressed. We have a couple comments that would improve the clarity or provide consistency in style including the following:

(1) The use of the term "invade" when describing ER entry into spines gives the impression that ER is not supposed to go in there. We much prefer the more neutral description e.g., "entry" or "visits" (which the authors already use in the manuscript) for this description. We like "visits" because it fits the activity- and ATP-dependent nature of ER trafficking into spines by Myo V.

We have replaced ‘invade’ and ‘invasion’ by synonyms as suggested.

(2) The author could improve the transition to the last paragraph of Results section by mentioning why they are looking at surface expression of AMPAR. Although it seems obvious, the current transition is a bit abrupt.

We inserted the following sentence:

“Electrophysiological induction of long-term plasticity assesses relative changes in synaptic strength, but provides little information about the absolute strength of synapses at baseline, e.g. the number of glutamate receptors.”

Finally, we were unable to find in the manuscript the author's response to reviewer #1's comment two -- regarding ChR-mediated activation of synapses.

We added the following sentence to the methods section:

“When ChR2 is expressed in CA3 pyramidal cells using AAV9, paired-pulse ratios are identical to electrophysiological stimulation, indicating physiological release probability (Jackman et al., 2014).”

Reviewer #3 (Remarks to the Author):

The authors have responded thoroughly to the comments of all previous reviewers, and have added clarifying new material to the manuscript. The manuscript is now ready for publication.

Thank you!